# A functional topography within the cholinergic basal forebrain for encoding sensory cues and behavioral reinforcement outcomes

**Blaise Robert[1], Eyal Y Kimchi[1,2], Yurika Watanabe[1], Tatenda Chakoma[1], Miao Jing[3], Yulong Li[4], Daniel B Polley[1,5]***

[1]Eaton-Peabody Laboratories, Massachusetts Eye and Ear Infirmary, Boston, United States; [2]Department of Neurology, Massachusetts General Hospital, Boston, United States; [3]Chinese Institute for Brain Research, Beijing, China; [4]State Key Laboratory of Membrane Biology, Peking University School of Life Sciences; PKU-IDG/McGovern Institute for Brain Research; Peking-Tsinghua Center for Life Sciences, Beijing, Beijing, China; [5]Department of Otolaryngology - Head and Neck Surgery, Harvard Medical School, Boston, United States

**Abstract** Basal forebrain cholinergic neurons (BFCNs) project throughout the cortex to regulate arousal, stimulus salience, plasticity, and learning. Although often treated as a monolithic structure, the basal forebrain features distinct connectivity along its rostrocaudal axis that could impart regional differences in BFCN processing. Here, we performed simultaneous bulk calcium imaging from rostral and caudal BFCNs over a 1-month period of variable reinforcement learning in mice. BFCNs in both regions showed equivalently weak responses to unconditioned visual stimuli and anticipated rewards. Rostral BFCNs in the horizontal limb of the diagonal band were more responsive to reward omission, more accurately classified behavioral outcomes, and more closely tracked fluctuations in pupil-indexed global brain state. Caudal tail BFCNs in globus pallidus and substantia innominata were more responsive to unconditioned auditory stimuli, orofacial movements, aversive reinforcement, and showed robust associative plasticity for punishment-predicting cues. These results identify a functional topography that diversifies cholinergic modulatory signals broadcast to downstream brain regions.

**\*For correspondence:** Daniel_Polley@meei.harvard.edu

**Competing interest:** The authors declare that no competing interests exist.

## Editor's evaluation

Cholinergic projections from the basal forebrain throughout the cortex are known to regulate arousal, signal transmission and plasticity. The basal forebrain shows distinct connectivity with the cortex along its anteroposterior axis that could entail distinct modulation of different parts of cortex. By performing long-term imaging of anterior and posterior basal forebrain activity during reinforcement learning, this study finds distinct reward, learning and sensory correlates of anterior and posterior basal forebrain activity, demonstrating that the cholinergic modulation of downstream cortical areas exhibits a functional topography.

## Introduction

Basal forebrain projections innervate the neocortex, hippocampus, and amygdala to regulate stimulus salience and global brain state across a wide range of timescales (for recent reviews, see *Disney*

*and Higley, 2020*; *Monosov, 2020*; *Sarter and Lustig, 2020*). The basal forebrain is not a mono-lithic structure, but rather a constellation of discrete brain areas that feature distinct combinations of neurochemical cell types and distinct arrangements of afferent and efferent connections (*Gielow and Zaborszky, 2017*; *Li et al., 2018*; *Rye et al., 1984*; *Zaborszky et al., 2012*). Any single region of the basal forebrain is composed of glutamatergic, GABAergic, and cholinergic neurons, which can each exhibit distinct downstream targeting and functional response properties (*Do et al., 2016*; *Laszlovszky et al., 2020*; *Yang et al., 2017*). As a whole, the basal forebrain is understood to contribute to learning, memory, attention, arousal, and neurodegenerative disease processes (*Everitt and Robbins, 1997*; *Monosov, 2020*; *Zaborszky et al., 2012*). However, the heterogeneity of cell types and projection targets have made it challenging to identify specific computations or specialized feature processing performed by 'the' basal forebrain, underscoring the need for cell type-specific recordings from targeted regions in task-engaged animals.

Basal forebrain cholinergic neurons (BFCNs), though numerically the rarest major neurochemical class of basal forebrain neuron (*Gritti et al., 2006*), are by far the most extensively studied. In rats and mice, where cholinergic neurons can be accessed for tracing, monitoring, and manipulation with transgenic approaches, BFCNs exhibit distinct arrangements of afferent and efferent connections along the extended rostrocaudal axis (*Gielow and Zaborszky, 2017*). BFCNs in rostral structures such as the horizontal limb of the diagonal band of Broca (HDB) feature strong reciprocal connectivity with prefrontal cortex and lateral hypothalamus, with additional projections to entorhinal cortex, olfactory bulb, and pyriform cortex (*Bloem et al., 2014*; *Gielow and Zaborszky, 2017*; *Li et al., 2018*; *Rye et al., 1984*; *Zaborszky et al., 2012*; *Figure 1A*). By contrast, BFCNs at the caudal tail of the basal forebrain, at the intersection of globus pallidus and substantia innominata (GP/SI), receive strong inputs from the caudate putamen, the medial geniculate, and posterior intrathalamic nuclei, and are the primary source of cholinergic input to the auditory cortex (ACtx), with comparatively weak projections to frontal cortical areas (*Chavez and Zaborszky, 2017*; *Guo et al., 2019*; *Kamke et al., 2005*; *Kim et al., 2016*; *Rye et al., 1984*; *Zaborszky et al., 2012*).

Although rostral and caudal BFCNs are wired into distinct anatomical networks, the suggestion is that they broadcast a relatively unified signal to downstream brain areas. The evidence for this conclusion primarily comes from two types of measurements. First, there are many converging reports of strong, short-latency BFCN responses to aversive stimuli such as air puffs or foot shock whether recordings are made from HDB (*Hangya et al., 2015*; *Harrison et al., 2016*; *Laszlovszky et al., 2020*; *Sturgill et al., 2020*), from the caudal extreme of the basal forebrain, GP/SI (*Guo et al., 2019*), or from an intermediate region of rodent SI often labeled as nucleus basalis (*Hangya et al., 2015*; *Laszlovszky et al., 2020*; *Letzkus et al., 2011*). Second, cortical fluorescence imaging of genetically encoded acetylcholine (ACh) sensors or calcium signals in BFCN axons have demonstrated a strong correspondence between cholinergic activity and behavioral indices of global arousal, as determined from EEG markers, isoluminant pupil diameter changes, and gross motor markers such as grooming or locomotion (ACh sensor imaging – *Lohani et al., 2021*; *Teles-Grilo Ruivo et al., 2017*; calcium imaging for HDB – *Harrison et al., 2016*; *Sturgill et al., 2020*; nucleus basalis – *Reimer et al., 2016*; GP/SI – *Nelson and Mooney, 2016*).

On the other hand, there are many inconsistencies in the emerging BFCN literature. These discrepancies could reflect differences in the anatomical source of BFCN activity, or they could arise from differences in mouse lines, behavioral task designs, and measurement techniques. For example, auditory cue-evoked BFCN responses have been described as absent altogether (*Hangya et al., 2015*), observed only for reward-predictive sounds (*Crouse et al., 2020*; *Harrison et al., 2016*; *Kuchibhotla et al., 2017*; *Parikh et al., 2007*), or enhanced after reinforcement learning but present even for unconditioned stimuli (*Guo et al., 2019*). Similarly, behavioral accuracy in discrimination tasks have been classified from BFCN activity both preceding and following the sensory cue (*Kuchibhotla et al., 2017*; *Parikh et al., 2007*), only from the post-cue response period (*Laszlovszky et al., 2020*; *Sturgill et al., 2020*), or only from putative non-cholinergic cell types (*Hangya et al., 2015*; *Lin and Nicolelis, 2008*). Reward-evoked BFCN activity has been described as weak overall (*Crouse et al., 2020*; *Harrison et al., 2016*; *Parikh et al., 2007*) or rapid and quite strong, particularly for uncertain rewards (*Hangya et al., 2015*; *Laszlovszky et al., 2020*; *Sturgill et al., 2020*; *Teles-Grilo Ruivo et al., 2017*). Finally, the relationship between BFCN activity and movement is unclear, with variable reports of strong recruitment by orofacial movements or locomotion occurring outside of a behavioral

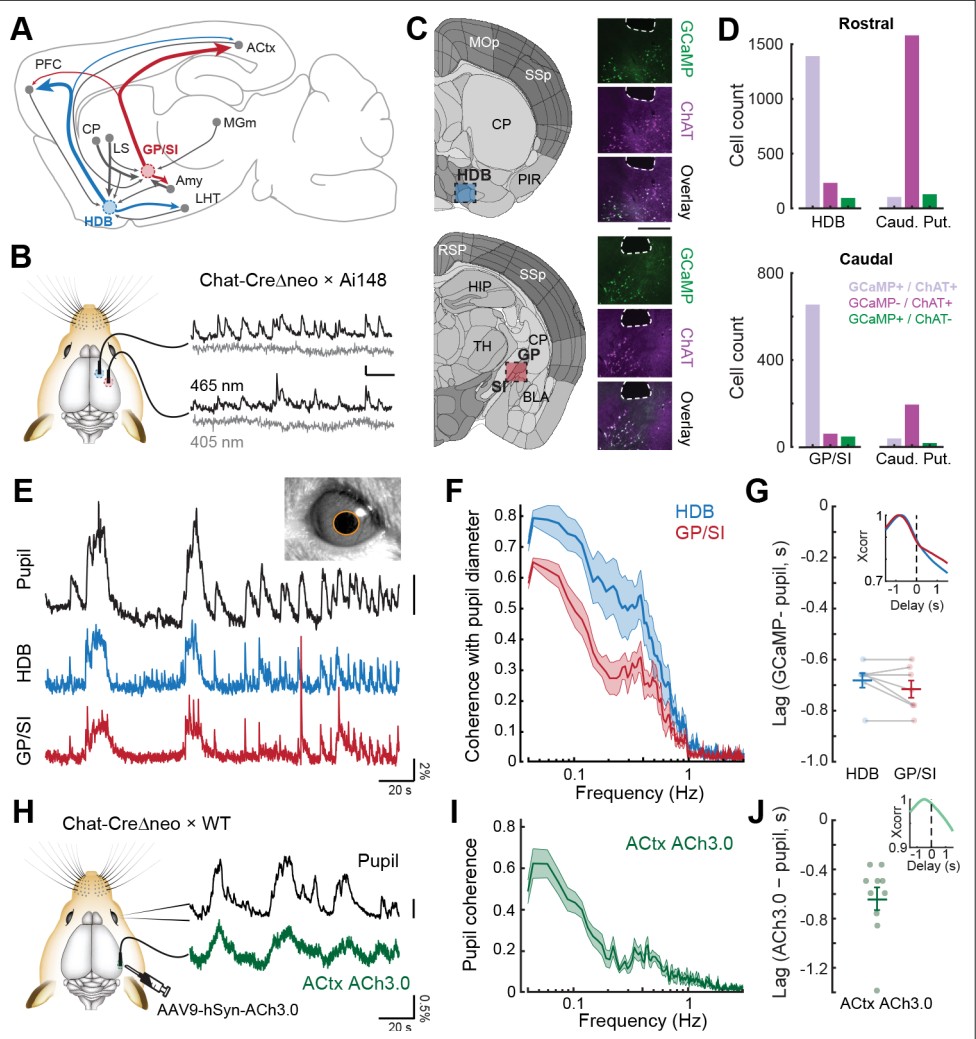

**Figure 1.** Bulk basal forebrain cholinergic neuron (BFCN) activity and cortical acetylcholine release closely correspond with pupil-indexed global brain state. (**A**) Mid-sagittal diagram of the mouse brain depicting the diversity in major inputs (gray) and outputs (colored) between a rostroventral basal forebrain structure, the horizontal limb of the diagonal band of Broca (HDB), and the caudodorsal tail of the basal forebrain, the boundary of the globus pallidus and substantia innominata (GP/SI). ACtx = auditory cortex, MGm = medial subdivision of the medial geniculate body, LHT = lateral hypothalamus, Amy = amygdala, LS = lateral septum, CP = caudate putamen, PFC = prefrontal cortex. (**B**) Dual bulk fiber-based calcium imaging from basal forebrain cholinergic neurons was performed from the HDB and GP/SI of ChAT-Cre-Δneo × Ai148 mice. Dual wavelength imaging allowed separate visualization of calcium-independent fluorescence (405 nm) from calcium-dependent fluorescence (465 nm). Vertical and horizontal scale bars reflect 1% DF/F and 5 s, respectively. (**C**) Coronal diagrams are adapted from the adult mouse coronal reference atlas created by the Allen Institute for Brain Science. Diagrams illustrate anatomical landmarks at the rostral (top) and caudal (bottom) imaging locations. Post-mortem fluorescence photomicrographs of brain sections immunolabeled for the ChAT protein depict the outline of the fiber path and the position of HDB, GP, and SI. GCaMP and ChAT fluorescence channels and their overlay to illustrate the strong co-localization of GCaMP in ChAT neurons within HDB and GP/SI regions near the fiber tip. Scale bar = 0.5 mm. (**D**) Cells from regions of interest below the fiber tip were counted based on their expression of GCaMP-only (green), ChAT-only (magenta), or both GCaMP and ChAT (lavender). The same analysis was performed on cells within the caudate putamen of the dorsal striatum. Numbers indicate the number of neurons in the corresponding category. (**E**) Isoluminous spontaneous pupil dilations in an example mouse were visualized in combination with GCaMP imaging from HDB and GP/SI. Pupil scale bar depicts a five pixel² areal change. (**F**) Mean ± SEM coherence of HDB and GP/SI GCaMP activity with pupil-indexed brain state in isoluminous conditions without any explicit environmental stimuli or task demands. N = 7 mice provided data for pupil, HDB, and GP/SI. Basal forebrain GCaMP signals closely track slow (<0.5 Hz) changes in pupil diameter, though the correspondence

*Figure 1 continued on next page*

*Figure 1 continued*

is stronger overall in HDB than in GP/SI (two-way repeated measures ANOVA, main effect for brain structure, F = 12.58, p = 0.01). (**G**) HDB and GP/SI GCaMP changes lead pupil fluctuations by approximately 0.7 s. *Inset:* Cross-correlation of the HDB and GP/SI GCaMP signals with pupil fluctuations. Individual data points depict the time value corresponding to the peak of the cross-correlograms from individual mice. Mean ± SEM values are provided at left and right. (**H**) Tapered fiber imaging of the ACh3.0 fluorescence during pupil videography. Scale bar depicts a five pixel diameter change. (**I**) Mean ± SEM coherence of ACtx ACh3.0 with pupil-indexed arousal in isoluminous conditions without any explicit environmental stimuli or task demands. N = 10 mice. Pupil coherence was qualitatively similar to GP/SI GCaMP coherence, which is expected on account of its stronger anatomical projection to ACtx. (**J**) ACtx ACh3.0 changes lead pupil fluctuations by approximately 0.6 s. *Inset:* Cross-correlation of the ACtx ACh3.0 signal with pupil fluctuations. Individual data points depict the time value corresponding to the peak of the cross-correlograms from individual mice. Mean ± SEM values are provided at left and right.

The online version of this article includes the following figure supplement(s) for figure 1:

**Source data 1.** Counts of GCaMP-expressing and ChAT-expressing cells in horizontal limb of the diagonal band of Broca (HDB), globus pallidus and substantia innominata (GP/SI), and the rostral and caudal caudate putamen.

**Figure supplement 1.** Anatomical locations of horizontal limb of the diagonal band of Broca (HDB) and globus pallidus and substantia innominata (GP/SI) fiber tips.

task (*Harrison et al., 2016*; *Nelson and Mooney, 2016*), strong only for movements associated with reinforcement (*Crouse et al., 2020*), or absent, whether movements were linked to reinforcement or not (*Hangya et al., 2015*; *Parikh et al., 2007*). In fact, while mesoscale imaging from the entire dorsal surface of the mouse neocortex was recently used to confirm an overall strong association between motor activity, global brain state, and ACh release, the findings also emphasized clear differences between behavioral states and spatiotemporal ACh dynamics, again suggesting functional heterogeneity in the sources of cholinergic input innervating anterior and posterior cortical regions (*Lohani et al., 2021*).

To better understand whether the disparate findings described above may reflect regional specializations for processing sensory and reinforcement signals within the cholinergic basal forebrain, we developed an approach to minimize inter-subject variation by testing all of the experimental features mentioned above in individual mice while making simultaneous fiber-based bulk GCaMP recordings from BFCNs in HDB and GP/SI. For some variables, we observed closely matched responses in rostral and caudal regions, suggesting a common output that would be broadcast to downstream brain regions. For example, both HDB and GP/SI exhibited equivalently weak overall responses to unconditioned visual stimuli and anticipated rewards. For other measures, we noted clear differences between BFCN activity in each region: HDB exhibited a comparatively strong association with pupil-indexed brain state, behavioral trial outcome, and with the omission of expected rewards. Response amplitudes for aversive stimuli were larger in GP/SI, as were responses to orofacial movements, unconditioned auditory stimuli, and learning-related enhancement of punishment-predicting auditory cues. These findings identify a coarse functional topography within the cholinergic basal forebrain that can be interpreted in light of the distinct connectivity of each region and will motivate future hypotheses about the causal involvement of each region in brain function and behavior.

## Results

### A transgenic strategy for selective GCaMP expression in HDB and GP/SI BFCNs

To characterize regional specializations within the cholinergic basal forebrain across a wide range of task-related variables, we performed dual fiber imaging from HDB and GP/SI in the right hemisphere of Chat-Cre mice that were crossed to the GCaMP6f reporter line, Ai148 (*Figure 1B–C*). Using cre-expressing mice for functional characterization of cholinergic neurons can be challenging. ChAT$_{(BAC)}$-Cre and ChAT$_{(IRES)}$-Cre homozygous mice exhibit behavioral irregularities that can be avoided by using ChAT$_{(IRES)}$-Cre hemizygous littermates (*Chen et al., 2018*). Ectopic expression in glia and non-cholinergic neurons can also be a problem, even in popular ChAT$_{(IRES)}$-Cre lines, either because the presence of a frt-flanked neo cassette can result in off-target expression, or because a fraction

of glutamatergic neurons express ChAT transiently during development and would therefore still be labeled with Cre-based transgenic expression approaches (*Nasirova et al., 2020*).

Here, we used hemizygous offspring from the ChAT(IRES)-CreΔneo line, in which the neo cassette is removed to reduce ectopic expression (*Nasirova et al., 2020*). We confirmed that GCaMP expression was almost entirely restricted to cholinergic neurons within the HDB and GP/SI by immunolabeling regions near the end of the fiber tips for ChAT in a subset of implanted mice (N = 4, see *Figure 1— figure supplement 1* for a presentation of all 22 fiber tip locations in 11 mice). ChAT-negative neurons that expressed GCaMP were rare, amounting to just 95/1719 in HDB (5.5%) and 48/764 in GP/SI (6.3%) (*Figure 1D*, left). As identified in prior studies, we observed aberrant expression in brain regions outside of the basal forebrain, including both the near-complete absence of GCaMP expression in ChAT+ striatal interneurons (*Figure 1D*, right) but also ectopic expression of GCaMP in ChAT-negative cells in neocortex and hippocampus. Therefore, while our transgenic strategy was appropriate for bulk imaging from cholinergic neurons in HDB and GP/SI cholinergic neurons (and in fact was aided by the absence of striatal GCaMP expression), it would not necessarily be a valid strategy for the study of other brain regions.

## Strong coherence between pupil-indexed arousal and cholinergic activity

Basal forebrain neurons have a well-established role in regulating global brain state (*Buzsaki et al., 1988*; *Kim et al., 2015*; *Yang et al., 2017*). The cholinergic basal forebrain, in particular, is a key regulator of neocortical excitability across sleep states as well as levels of vigilance during quiescent awake states (*Buzsaki et al., 1988*; *Everitt and Robbins, 1997*; *McGinley et al., 2015b*; *Reimer et al., 2016*; *Teles-Grilo Ruivo et al., 2017*). Under isoluminous lighting conditions, pupil diameter provides a sensitive index of arousal and has been shown to co-vary with GCaMP activity measured in cholinergic basal forebrain axon fields within the neocortex (*Nelson and Mooney, 2016*; *Reimer et al., 2016*). Prior measurements were either made in ChAT-Cre× GCaMP reporter lines or via relatively large viral solution injection quantities (0.4–1 µL), which leaves unresolved the question of how the activity of cholinergic neurons in specific regions of the basal forebrain corresponds to pupil-indexed arousal state. To address this point, we simultaneously monitored spontaneous pupil fluctuations alongside fiber-based GCaMP imaging from HDB and GP/SI. We observed a striking correspondence between spontaneous pupil dilations and slow fluctuations in GCaMP signal amplitudes in both regions of the cholinergic basal forebrain (*Figure 1E*). GCaMP coherence with pupil fluctuations was significantly higher in HDB than GP/SI, where bulk calcium dynamics could account for as much as 80% of the variability in slow pupil changes (*Figure 1F*, statistical reporting provided in figure legends). The timing of correlated GCaMP transients and pupil dilations were similar across brain areas, where GCaMP signals led pupil dilations by approximately 0.7 s (*Figure 1G*).

One of the underlying assumptions in our approach is that bulk calcium imaging from ChAT-Cre neurons in the basal forebrain is a useful way to measure the suprathreshold activity of local BFCNs and infer the timing of ACh release in downstream targets. For example, based on the correspondence between basal forebrain bulk GCaMP levels and pupil diameter, it would be reasonable to hypothesize that ACh levels also co-vary with pupil dilations with a similar coherence. HDB and GP/SI BFCNs both project to ACtx, although BFCN → ACtx projections are far more numerous in GP/SI than HDB (*Chavez and Zaborszky, 2017*; *Guo et al., 2019*; *Kamke et al., 2005*; *Rye et al., 1984*). To monitor ACh dynamics in ACtx related to pupil fluctuations, we expressed the genetically encoded ACh fluorescent sensor, GRAB$_{ACh}$3.0 (ACh3.0), in ACtx neurons and monitored fluorescence dynamic with tapered optical fibers (*Figure 1H*; *Jing et al., 2020*; *Pisano et al., 2019*). As expected, coherence between ACtx ACh3.0 fluorescence and pupil fluctuations strongly resembled GCaMP coherence from GP/SI cell bodies, both in terms of the strong coherence with slow (<0.1 Hz) changes in pupil diameter (*Figure 1I*) and in terms of timing, where ACh3.0 signal surges led pupil dilations by approximately 0.6 s (*Figure 1J*). These findings validate our use of bulk fiber-based calcium imaging in the GCaMP reporter line as a useful way to monitor cholinergic basal forebrain activity and additionally demonstrate a strong correspondence between pupil-indexed arousal and activity surges in HDB and – to a lesser extent – GP/SI.

## Audiovisual stimulus encoding and habituation across the cholinergic basal forebrain

Having confirmed that our dual fiber bulk GCaMP imaging approach could capture the expected relationship between pupil-indexed brain state and cortical ACh levels, we next tested regional variations in BFCN responses for passively presented unconditioned auditory and visual stimuli that had no explicit behavioral significance (*Figure 2A*). As illustrated in an example mouse, presentation of novel – but behaviorally irrelevant – drifting visual gratings elicited weak responses from both regions. Auditory spectrotemporal gratings (i.e., ripples) elicited comparable responses in HDB but robust responses in GP/SI even at the lowest sound levels tested (*Figure 2B*). Quantification of visual- and sound-evoked responses across all mice (N = 11) confirmed modest bulk BFCN responses to visual gratings of varying contrast that did not differ significantly between HDB and GP/SI (*Figure 2C*, top). BFCN responses to unconditioned auditory stimuli were markedly different than visual stimuli, as observed for both complex broadband ripple sounds (*Figure 2C*, middle) and brief pure tone pips (*Figure 2C*, bottom). In GP/SI, significant BFCN responses were observed for both types of sounds at all stimulus intensities and were all significantly greater than the corresponding HDB responses.

To better understand how modest HDB and robust GP/SI responses to broadband auditory ripples related to stimulus novelty and stimulus-elicited arousal, we returned to an analysis of pupil dilations, which can be elicited by sounds that are novel, emotionally evocative, or require heightened listening effort (*Ebitz and Moore, 2018*; *McGinley et al., 2015b*; *Zekveld et al., 2018*). Along these lines, we observed large pupil dilations to the first presentation of an auditory ripple at 70 dB SPL, which then habituated to approximately 50% of their initial amplitude after one or two trials, presumably reflecting the loss of stimulus novelty (*Figure 2D*). Ripple-evoked BFCN responses decayed in parallel with pupil responses, where responses decreased by approximately 30% after the first presentation before stabilizing at approximately 60% of the initial amplitude across subsequent presentations. Although the ripple-evoked response amplitude was greater overall in GP/SI than HDB, the proportional decay with habituation was equivalent (*Figure 2E*). Rapid habituation of BFCN responses was also observed for auditory ripples presented at lower sound levels, visual gratings at lower contrast, and for moderate intensity pure tones, providing further evidence that BFCN sensory responses were modulated stimulus novelty across a wide range of physical stimulus types (*Figure 2—figure supplement 2*). Finally, to control for the possibility that the progressive response decay reflected photobleaching of the sample or another source of measurement noise, we also quantified the amplitude of spontaneous GCaMP transients measured during trials in which neither auditory nor visual stimuli were presented. We found that the amplitude of spontaneous GCaMP transients was unchanged throughout the recording period, confirming that the reduced sensory-evoked GCaMP responses over the test session reflected habituation to stimulus novelty (*Figure 2F*).

## Stable BFCN responses to reward-predicting cues

Prior studies have described enhanced BFCN responses to sensory cues associated with reward (*Crouse et al., 2020*; *Harrison et al., 2016*; *Kuchibhotla et al., 2017*; *Parikh et al., 2007*) and co-modulation of BFCN activity rates with behavioral performance accuracy in sensory detection and recognition tasks (*Kuchibhotla et al., 2017*; *Laszlovszky et al., 2020*; *Parikh et al., 2007*; *Sturgill et al., 2020*). To determine how BFCN activity dynamics related to appetitive learning and task performance, we conditioned mice to lick a delivery spout shortly following the onset of a tone to receive a sugar water reward (*Figure 3A*). To temporally separate the cue, operant motor response, and reinforcement timing, the reward was delayed until mice produced an extended, vigorous bout of licking (≥7 licks in 2.8 s). Although the rates of procedural learning varied somewhat between mice (*Figure 3B*), all mice learned the task within a few sessions and either detected the tone to receive reward (hit) or failed to lick at all in response to the tone (miss), with very few instances of partial hits (>0 but <7 licks in 2.8 s) observed after the first few behavioral sessions (*Figure 3C*).

We contrasted BFCN activity on hit and miss trials over the course of operant testing in HDB (*Figure 3D*) and GP/SI (*Figure 3E*). On average, tone-evoked responses were not greatly changed late in training, after mice had learned the stimulus-reward association (*Figure 3F–G*). Responses were slightly elevated at longer latencies after stimulus onset early in training, though this difference could be explained by differences in lick rate duration over the course of training (*Figure 3—figure supplement 1*). Overall, BFCN responses to reward-predicting tones did not significantly change over the

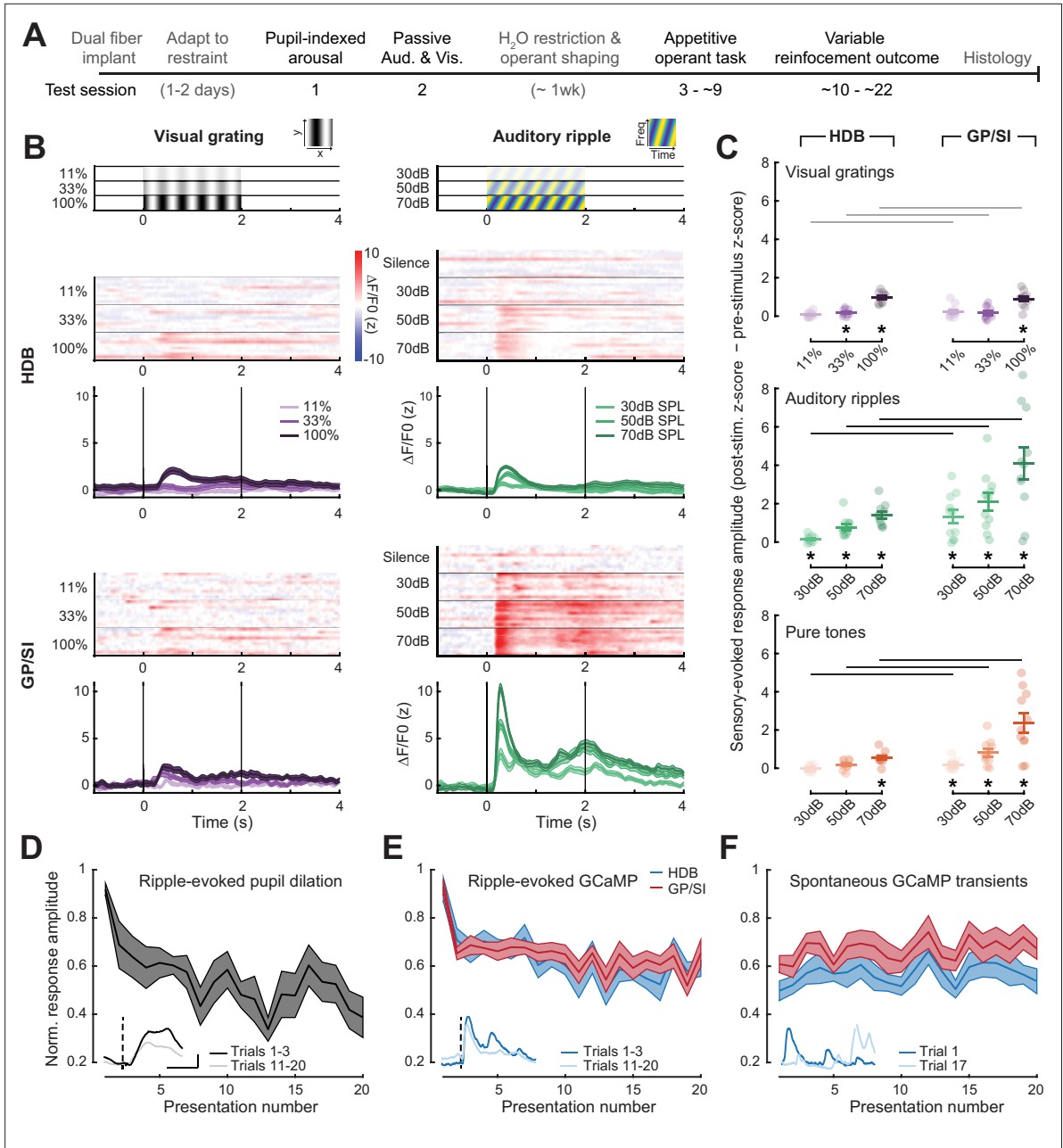

**Figure 2.** Strong, rapidly habituating responses to unconditioned auditory - but not visual – stimuli in globus pallidus and substantia innominata (GP/SI) cholinergic neurons. (**A**) Timeline for measurement sessions (black text) and procedures (gray text) performed in each of 11 ChAT-Cre-Δneo × Ai148 mice. Basal forebrain cholinergic neuron (BFCN) responses to unconditioned auditory and visual stimuli described below were measured during test session 2. (**B**) BFCN responses to drifting visual gratings of varying contrast (left) and auditory spectrotemporal ripples of varying sound levels (right) are shown for an example mouse. Heatmaps depict fractional change values for individual trials in horizontal limb of the diagonal band of Broca (HDB) (top row) and GP/SI (bottom row). Line plots depict mean ± SEM z-scored fractional change across all trials. Vertical bars denote onset and offset of the 2 s stimulus period. (**C**) Evoked response amplitudes to auditory and visual stimuli in HDB (*left column*) and GP/SI (*right column*). Circles denote individual mice (N = 11 for all conditions), bars denote sample mean and SEM sensory response amplitudes. Responses at variable stimulus intensities are averaged across horizontal/vertical visual orientations (*top*), upward and downward auditory frequency modulation (*middle*), and low, middle, and high auditory pure tone frequencies (*bottom*). Refer to *Figure 2—figure supplement 1* for a comparison of responses to each direction of visual and auditory stimulus change. Sensory-evoked cholinergic responses to visual gratings and auditory ripples increase with intensity and contrast, but are stronger overall in GP/SI, particularly in the auditory modality (three-way repeated measures ANOVA with structure, stimulus level, and modality as independent variables: main effect for structure, F = 10.09, p = 0.01; main effect for stimulus level, F = 63.52, p = $2 \times 10^{-9}$; main effect for modality, F = 20.83, p = 0.001; modality × structure × level interaction term, F = 9.1, p = 0.002). Asterisks denote a significant difference in the peak post- and pre-stimulus response (paired t-test, p < 0.05, corrected for multiple comparisons). Black and gray horizontal bars denote significant and non-significant

*Figure 2 continued on next page*

*Figure 2 continued*

differences, respectively, in sensory-evoked response amplitudes between HDB and GP/SI (paired t-test, p < 0.05, corrected for multiple comparisons). (**D**) Mean ± SEM normalized pupil dilations evoked by 70 dB SPL auditory ripples significantly decreased over 20 presentations (one-way repeated measures ANOVA, F = 2.85, p = 0.0003; N = 7 mice). *Inset*: Mean sound-evoked pupil diameter change in an example mouse for trials 1–3 vs. 11–20. Inset scale bar = 1 z-score and 2 s and applies to all inset panels below. Vertical dashed line = onset of the 2 s stimulus. (**E**) Mean ± SEM normalized BFCN response to auditory ripples were significantly and equivalently reduced in HDB and GP/SI over 20 presentations (two-way repeated measures ANOVA with structure and presentation number as independent variables: main effect for structure, F = 0.51, p = 0.49; main effect for presentation number, F = 6.11, p = 5 × 10$^{-12}$; N = 11 mice). *Insets*: Mean response from an HDB fiber of an example mouse for trials 1–3 vs. 11–20. *Figure 2—figure supplement 2* presents habituation functions for other auditory and visual stimulus types at varying stimulus intensities. (**F**) Mean ± SEM normalized BFCN spontaneous GCaMP transient amplitudes did not change over 20 measurement blocks (two-way repeated measures ANOVA with structure and presentation number as independent variables: main effect for structure, F = 0.80, p = 0.70; presentation number × structure interaction term, F = 0.57, p = 0.93; N = 11 mice). *Insets*: Spontaneous transients from an HDB fiber in two trials for which no stimulus was presented.

The online version of this article includes the following figure supplement(s) for figure 2:

**Figure supplement 1.** Equivalent basal forebrain cholinergic neuron (BFCN) responses to varying directions of auditory and visual drifting gratings.

**Figure supplement 2.** Basal forebrain cholinergic neuron (BFCN) responses to unconditioned sensory cues rapidly habituate across stimulus type, modality, and intensity.

course of learning for hit or miss trials in either brain area (***Figure 3H***). This result stands in contrast to prior reports of enhanced responses for sounds with a learned reward association, though it should be noted none of these prior studies had targeted BFCNs in HDB or GP/SI (***Crouse et al., 2020***; ***Harrison et al., 2016***; ***Kuchibhotla et al., 2017***; ***Parikh et al., 2007***). Another possibility is that response enhancement to reward-predicting sounds had already occurred during the initial shaping period that preceded the first operant imaging session, thereby escaping our analysis. Although performance in the Go-NoGo auditory task clearly improved over the course of our imaging period (***Figure 3B–C***), learning-related enhancements of cue-evoked BFCN responses can occur within just a few behavioral sessions (***Crouse et al., 2020***; ***Sturgill et al., 2020***), so we cannot rule out this possibility.

## BFCN activity preceding and following cue onset predicts behavioral trial outcome

Although cue-evoked response amplitudes were not obviously changed over the course of rewarded learning, they clearly differed between hit and miss trials. Cue-evoked responses were strongly reduced in HDB and GP/SI on miss trials (***Figure 3D–H***), although this difference is confounded by the potential contribution of lick-related motor activity that would only occur on hit trials. For this reason, differences in the pre-cue baseline activity levels are particularly illuminating, as they can reveal associations between population BFCN activity and behavioral performance without the influence of task-related sensory inputs or movements. We found that mean BFCN activity measured in a 1 s period prior to cue onset was significantly elevated on miss trials in both structures, though the difference was significantly greater in HDB (***Figure 3I***).

To determine whether these differences were sufficient to classify single trial outcomes, we trained a decoder on bulk BFCN activity measured in the HDB fiber, the GP/SI fiber, or from the simultaneous activity from both fibers. This was accomplished by first reducing the dimensionality of the data matrix with principal components analysis and then training a binary support vector machine (SVM) on the principal components projection to classify whether the pre-cue (***Figure 4A***) or post-cue (***Figure 4C***) BFCN activity from a single trial culminated in a hit or miss outcome. Despite the limited spatial and temporal resolution of GCaMP fiber imaging, differences in both pre- and post-cue BFCN activity supported classification of behavioral trial outcome with an accuracy that was significantly greater than a randomized control assignment for all brain structures (***Figure 4B and D***). For either pre- and post-cue activity, the HDB fiber classification accuracy was significantly higher than GP/SI and was not significantly different than the combined activity from both fibers.

## Movement-related activity in the cholinergic basal forebrain

The results presented thus far identify clear functional differences between rostral and caudal BFCNs. We have shown that rostral HDB activity is more closely related to global brain state and behavioral accuracy, whereas the caudal GP/SI exhibited more robust responses to auditory stimuli, regardless of their novelty or behavioral relevance. As a next step, we investigated differential recruitment of each

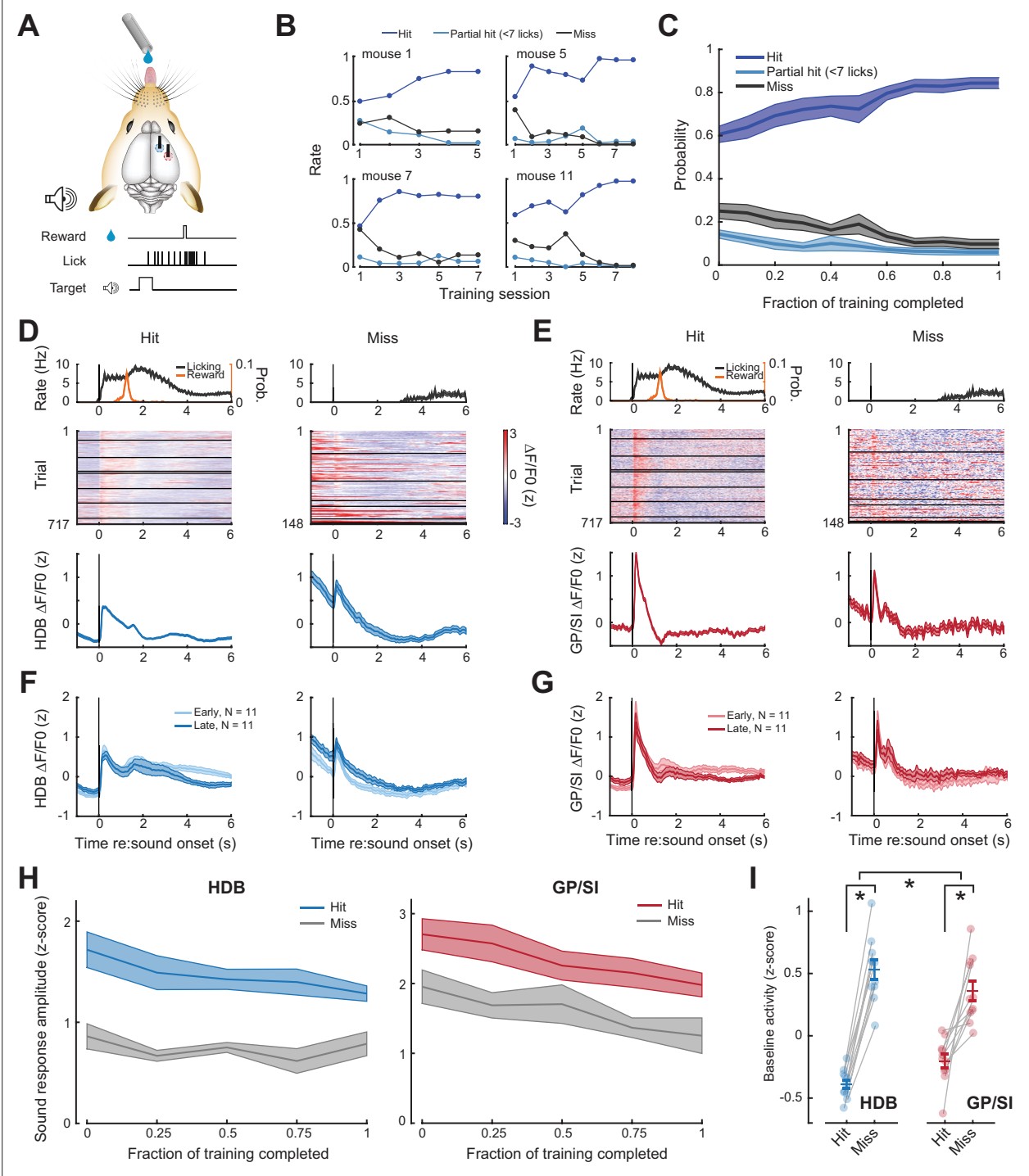

**Figure 3.** Pre-stimulus cholinergic basal forebrain activity distinguishes behavioral hit and miss trials during an auditory detection task. (**A**) Mice were rewarded for producing a vigorous bout of licking (at least 7 licks in 2.8 s) shortly after a low-, mid-, or high-frequency tone. (**B**) Learning curves from four example mice that became competent in the detection task at slightly different rates. (**C**) Mean ± SEM probability of hit, partial hit, and miss trial outcome as fraction of training completed in N = 11 mice. (**D–E**) Tone-evoked cholinergic GCaMP responses from the horizontal limb of the diagonal band of Broca (HDB) (**D**) and globus pallidus and substantia innominata (GP/SI) (**E**) of a single mouse from 717 hit and 148 miss trials distributed over eight appetitive conditioning sessions. Left columns present the timing of lickspout activity, reward probability, heatmaps single trial fractional change values, and mean ± SEM fractional change values. Right columns present the same data on miss trials. Horizontal black lines in heatmaps denote different daily recording sessions. Vertical lines denote tone onset. (**F–G**) Plotting conventions match D–E, except that data are averaged across all mice (N = 11) and the first third of training trials (early) are plotted separately from the last third of training trials (late). Training-related changes in the sensory-

*Figure 3 continued on next page*

*Figure 3 continued*

evoked responses were not observed, though see *Figure 3—figure supplement 1* for an analysis of small differences in the sustained response. (**H**) Mean ± SEM sound-evoked response amplitudes in all 11 mice were calculated by subtracting the mean activity during a 2 s pre-stimulus baseline period from the peak of activity within 400 ms of sound onset. Each behavior session was assigned to one of five different discrete time bins according to the fraction of total training completed. Although sound-evoked responses are reduced on miss trials compared to hit trials, they remain relatively stable across all conditions as mice learn to associate neutral sounds with reward (three-way repeated measures ANOVA with training time, trial type, and structure as independent variables: main effect for training time, F = 2.46, p = 0.08; main effect for trial type, F = 14.74, p = 0.012; training time × trial type × structure interaction, F = 0.56, p = 0.7). (**I**) Mean baseline activity during a 1 s period preceding stimulus onset on hit and miss trials. Circles denote individual mice (N = 11 for all conditions), bars denote sample mean and SEM. Pre-stimulus baseline activity was significantly higher on miss trials than hit trials, particularly in HDB (two-way repeated measures ANOVA with trial type and structure as independent variables: main effect for trial type, F = 102.04, p = 1 × 10$^{-6}$; trial type× structure interaction, F = 7.89, p = 0.02). Asterisks denote significant differences based on within-structure post hoc pairwise contrasts (p < 0.001 for both) or the trial type × structure interaction term (p = 0.2).

The online version of this article includes the following figure supplement(s) for figure 3:

**Figure supplement 1.** Lick rates may account for subtle differences in sustained basal forebrain cholinergic neuron (BFCN) sustained activity across learning.

region by motor activity by analyzing BFCN activity surrounding licking events during the inter-trial interval.

Licking behavior during the inter-trial period ranged from spurious checks of the lickspout, composed of just 1 or 2 successive licks, all the way to the occasional presentation of the operant lick bout behavior (i.e., a false alarm). As illustrated in an example mouse, we noted a modest increase in BFCN activity beginning shortly after the onset of an intense lick bout in GP/SI and, to a lesser extent, HDB (*Figure 5A*, left column). We also observed an unexpected second increase in BFCN activity following the offset of the lick bout (*Figure 5A*, right column). BFCN responses to the onset of licks increased monotonically across lick bout duration and, while fairly modest overall (i.e., when compared to cue-evoked responses), were significantly greater in GP/SI than HDB (*Figure 5B*). False alarm events during the inter-trial interval were uncommon overall, mostly occurring mid-way through the operant learning task (*Figure 5—figure supplement 1A*). Unlike the elevated BFCN activity prior to cue onset in undetected miss trials (*Figure 3I*), we did not observe a commensurate elevation in BFCN activity prior to false alarm events, suggesting that changes in baseline activity levels are more closely related to perceptual accuracy than behavioral action (*Figure 5—figure supplement 1B*).

We also noted phasic responses at the cessation of licking, but only when lick bouts exceeded the threshold for a false alarm event (*Figure 5C*). In GP/SI, we noted only a minimal response to the offset of ≥7 licks, which was not significantly greater than the response to shorter lick bouts. In HDB, which exhibited comparatively weak responses to movement onset, we observed significantly greater responses at the offset of lick bouts, but only when ≥7 licks were produced (*Figure 5D–E*). One interpretation of these findings is that the mouse occasionally deployed the full operant lick behavior during the silent inter-trial interval in anticipation of reward. In this scenario, phasic responses at the offset of false alarm events may reflect a reward omission response. Both cases – the increasing GP/SI activity with lick number and selective HDB responses after the omission of an anticipated reward - corroborate recent findings that BFCNs are more strongly recruited by motor actions that are expected to result in reward (*Crouse et al., 2020*), a possibility that we address more directly in the next stage of behavioral experiments.

## BFCN responses to punishment, reward, and reward omission

To address how behavioral reinforcement – and the omission of anticipated reinforcement – was related to activity in different regions of the BFCN, mice were advanced to the next phase of the operant training procedure, in which one of the tone frequencies maintained its association with reward, while the other two frequencies were either switched to reward omission or punishment (*Figure 6A*). Operant 'Go' responses (≥7 licks in 2.8 s) were initially high to all tone frequencies following the abrupt change in reinforcement outcome (*Figure 6B*). Within a few behavioral sessions, Go responses to the tone associated with a neutral outcome were reduced to approximately 40% of trials and Go response to the tone associated with tongue shock was only observed on approximately 25% of trials (*Figure 6C*).

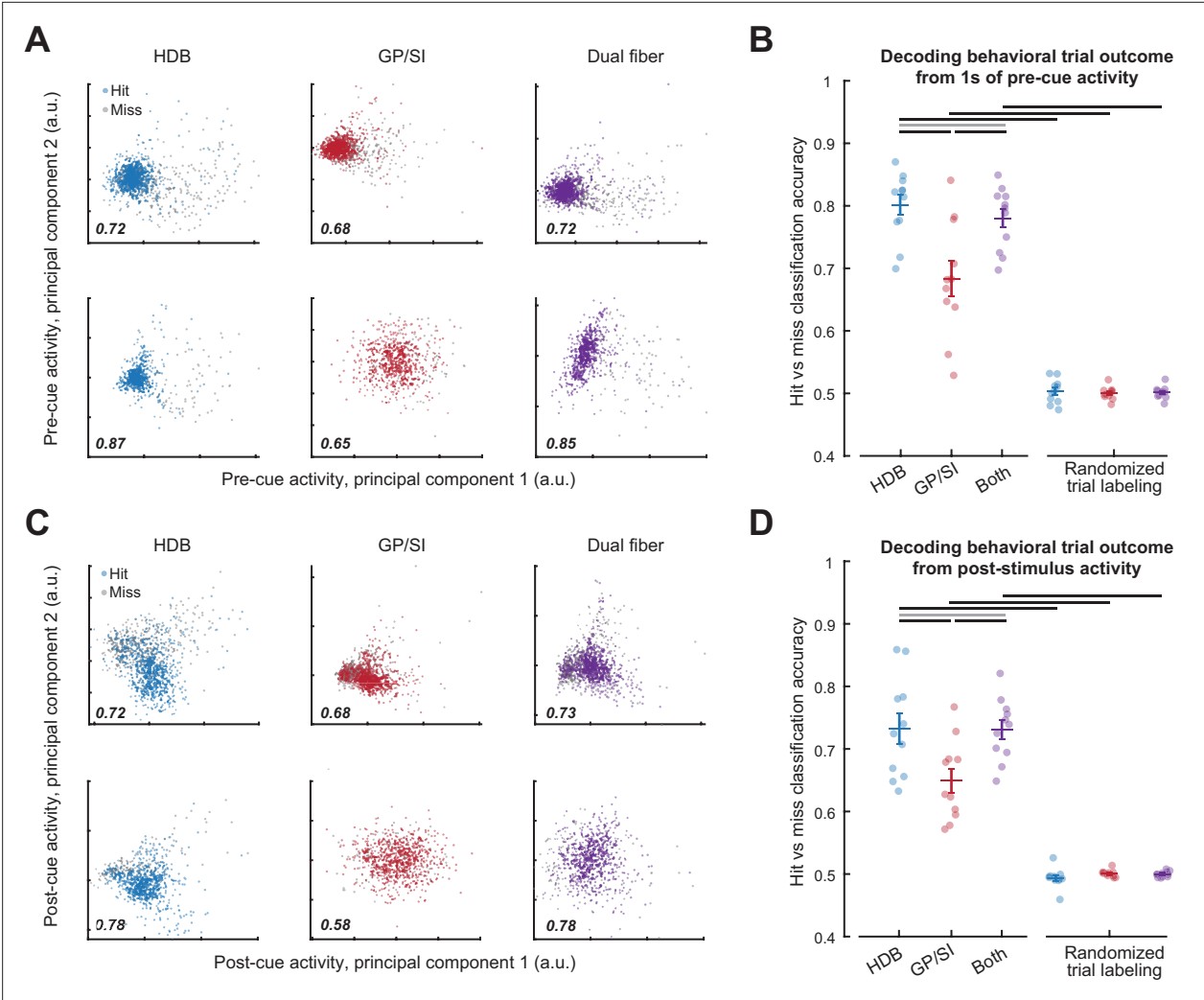

**Figure 4.** Pre- and post-cue basal forebrain cholinergic neuron (BFCN) activity predicts behavioral accuracy. (**A**) Bulk BFCN activity measured 1 s prior to tone onset for two representative mice. Circles denote activity from individual hit and miss trials projected onto the first two principal components. A support vector machine was used to assign principal component projections for individual trials to hit and miss outcomes. Classification accuracy is provided as the fraction of correctly assigned individual trials for horizontal limb of the diagonal band of Broca (HDB), globus pallidus and substantia innominata (GP/SI), and the simultaneous activity measured from both fibers (blue, red, and purple, respectively). (**B**) Accuracy for support vector machine classification of behavioral trial outcome based on 1 s of activity immediately preceding cue onset. Circles denote mean accuracy for the HDB, GP/SI, or both fibers in each individual mouse. Bars denote mean ± SEM. Baseline HDB activity more accurately decodes forthcoming trial outcome than GP/SI and is no worse than both fibers combined, though all conditions are significantly above chance (two-way repeated measures ANOVA with randomization and structure as independent variables: main effect for randomization, $F = 339.37$, $p = 5 \times 10^{-9}$; main effect for structure, $F = 11.64$, $p = 0.0004$). Black and gray horizontal lines indicate significant ($p < 0.01$ for all) and non-significant ($p = 0.05$) pairwise contrasts, respectively, after correcting for multiple comparisons. (**C**) Plotting conventions match A, except that data come from the 400 ms period immediately following cue onset. (**D**) Plotting conventions match B, except that data come from the 400 ms period immediately following cue onset. Post-cue HDB activity is less accurate at decoding forthcoming trial accuracy overall than baseline activity, though accuracy is still greater than chance and still relatively better in HDB than GP/SI (three-way repeated measures ANOVA with activity period, randomization, and structure as independent variables: main effect for activity period, $F = 10.57$, $p = 0.009$; main effect for randomization, $F = 339.37$, $p = 5 \times 10^{-9}$; main effect for structure, $F = 11.6$, $p = 4 \times 10^{-4}$). Black and gray horizontal lines indicate significant ($p < 0.04$ for all) and non-significant ($p = 0.96$) pairwise contrasts, respectively, after correcting for multiple comparisons.

This arrangement allowed us to contrast BFCN responses in HDB and GP/SI elicited by reward delivery, reward omission, and punishment (*Figure 6D*). We observed that BFCN responses to anticipated rewards were very weak in both HDB and GP/SI (*Figure 6E*). The omission of an anticipated reward was associated with a moderate response in HDB that was significantly greater than both reward delivery response from the same fiber and the reward omission response in GP/SI. Delivery of silent, noxious stimulus elicited the strongest BFCN responses in both regions, although the response

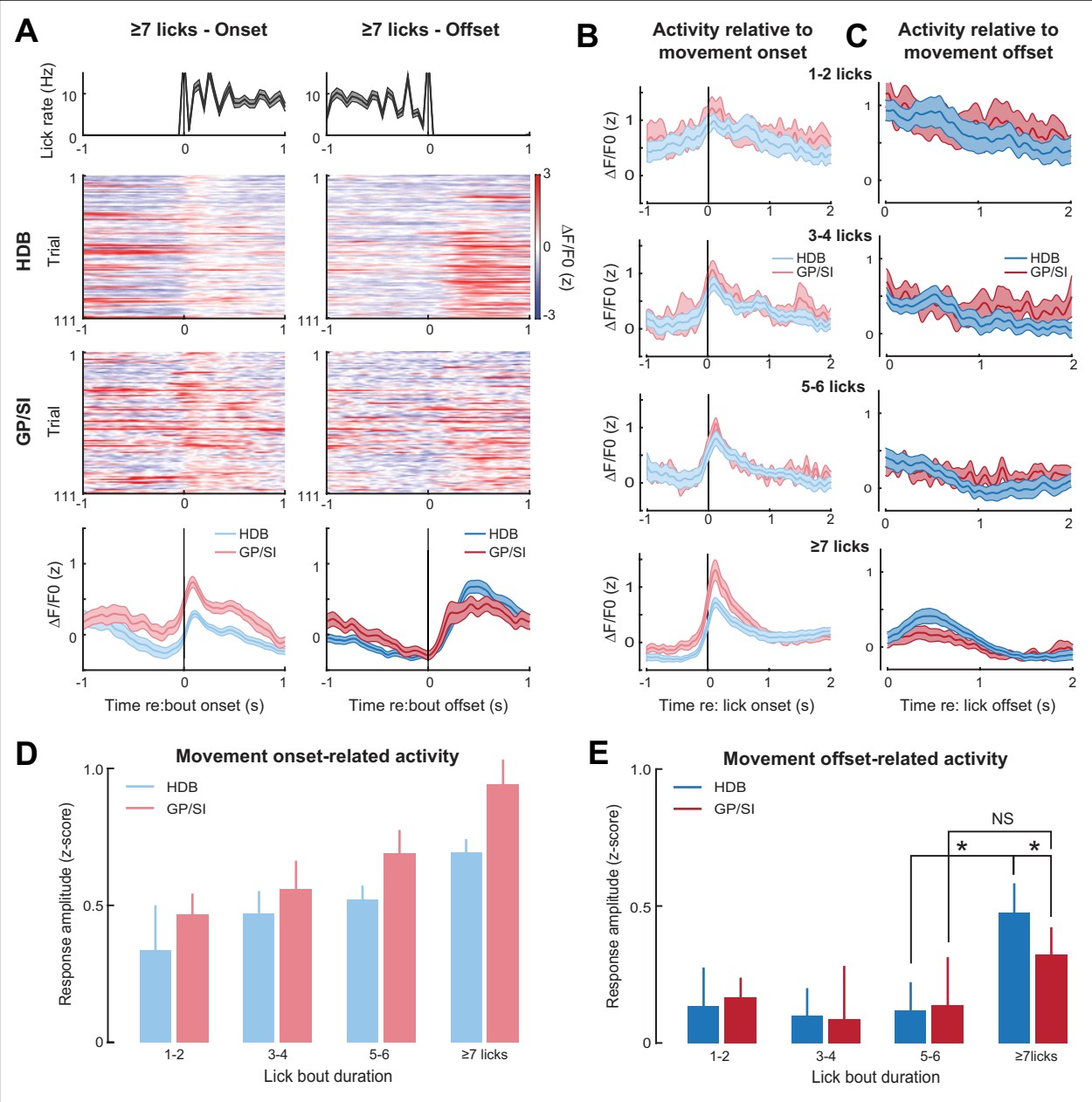

**Figure 5.** Motor-related activation of the cholinergic basal forebrain. (**A**) Horizontal limb of the diagonal band of Broca (HDB) and globus pallidus and substantia innominata (GP/SI) activity from an example mouse related to the onset (left column) and offset (right column) of vigorous lick bouts during the inter-trial period of the appetitive operant task. Line plots in top and bottom row reflect mean ± SEM. (**B–C**) Inter-trial lick bouts were binned according to whether they contained 1–2, 3–4, 5–6, or the full 7+ licks that would have triggered reward delivery if produced at the appropriate time during the operant task. Mean ± SEM activity from N = 11 mice related to the onset (**B**) or offset (**C**) of different lick bout durations. (**D**) Response amplitudes related to lick bout onset were calculated by subtracting the maximum activity from the 250 ms period preceding bout onset from the maximum activity occurring within 700 ms following lickspout contact. Movement-related responses increased with lick bout duration and were greater overall in GP/SI than HDB (two-way repeated measures ANOVA with bout duration and structure as independent variables: main effect for bout duration, F = 6.92, p = 0.001; main effect for structure, F = 6.33, p = 0.03). (**E**) Response amplitudes related to lick bout offset were calculated by subtracting the maximum activity from the 400 ms period preceding lick bout offset from the maximum activity occurring within 700 ms following lick spout offset. Overall, the offset of licking did not elicit a response (two-way repeated measures ANOVA with bout duration and structure as independent variables: main effect for bout duration, F = 1.47, p = 0.24). In HDB, a response was observed at the offset of licking, but only for intense bouts of ≥7 licks (pairwise post hoc contrast: 7+ vs. 5–6, p = 0.01). No comparable response was observed in GP/SI (pairwise post hoc contrast: 7+ vs. 5–6, p = 1; 7+ GP/SI vs. HDB, p = 0.03). Asterisks denote pairwise contrast p-values < 0.05 after correcting for multiple comparisons. NS = not significant.

The online version of this article includes the following figure supplement(s) for figure 5:

*Figure 5 continued on next page*

*Figure 5 continued*

**Figure supplement 1.** False alarms were relatively uncommon and were not associated with elevated baseline basal forebrain cholinergic neuron (BFCN) activity.

to shock was significantly greater in GP/SI than HDB (*Figure 6E*). BFCN response latencies to reward omission were significantly slower than other reinforcement types (mean ± SEM for omission vs. reward and shock; 1.04 ± 0.03 vs. 0.63 ± 0.03 s, for HDB and GP/SI, respectively; *Figure 6F*). The timing of the reward omission response was more precisely locked to lick bout offset than to timing of when reward would have been delivered. However, the response is not likely due to movement per se, because activity levels following lick bout cessation were significantly greater on reward omission trials than on trials when the reward was delivered and consumed (*Figure 6—figure supplement 1*). Recordings from unidentified basal forebrain neuron types in primates demonstrate that reward-omission responses occur only in a sub-type of neurons with slower, ramping responses (*Zhang et al., 2019*). Our observation of slower developing omission responses supports prior descriptions of reward timing and reinforcement prediction error encoding in BFCNs (*Chubykin et al., 2013*; *Sturgill et al., 2020*).

## Learning-related enhancement of BFCN responses to punishment-predicting cues

Our earlier work used a Pavlovian trace conditioning paradigm to identify a transient, selective enhancement of GP/SI BFCN activity to sounds associated with delayed aversive reinforcement. Enhanced BFCN single unit spiking emerged within minutes of pairing sound with aversive air puffs, while a slower, persistent enhancement of cue-evoked fiber-based GCaMP responses emerged 1 day after the initial pairing of sounds with foot shock to 'fill in' the silent gap separating the auditory cue and the delayed aversive reinforcement (*Guo et al., 2019*). Here, we did not observe enhancement of BFCN responses to reward-predictive cues (*Figure 3H*). To reconcile these findings with our prior study, we next examined whether auditory cues predicting aversive stimuli were enhanced after a reversal in reinforcement outcome.

When compared with the Phase 1 all-rewarded stage of the operant task, cue-evoked responses in HDB remained relatively constant over the remainder of conditioning, showing no significant differences between reward-related, omission-related, or punishment-related cues (*Figure 7A*). In GP/SI, responses to the tone frequencies associated with reward and reward omission were also relatively stable, but cue-evoked responses for the punishment-predicting tone frequency were enhanced within a few testing sessions following the change in reinforcement outcome (*Figure 7B*). These data confirm that sound-evoked responses are not changed for tone frequencies associated with anticipated reward or the unanticipated omission of reward (*Figure 7C*, left and middle). By contrast, cue-evoked responses increased by approximately 150% in GP/SI as the animal learned a new association between sound and punishment (*Figure 7C*, right).

As a final analysis that plays to the strength of the long-term fiber imaging approach, we concatenated the tone-evoked HDB and GP/SI responses across hundreds of trials – from the initial presentation day to the final operant behavioral session (639 presentations of a given tone frequency, on average; *Figure 7D*). When first exposed to pure tone stimuli on the initial passive characterization day, GP/SI BFCNs exhibited significantly greater within-session response habituation than HDB (*Figure 7E*, see also *Figure 2—figure supplement 2C*). Response habituation was reduced as mice became more familiar with the stimuli and task demands, such that tones associated with reward or reward omission showed stable levels of reduced habituation throughout Phases 1 and 2 of the operant task (*Figure 7F*). Interestingly, strong within-session habituation was rekindled later in training, though only in GP/SI and only for the tone frequency that was remapped to punishment (*Figure 7G*). Taken as a whole, these findings suggest that strong, rapidly habituating responses in the caudal BFCN may reflect the neural evaluation of potentially threatening stimuli.

## Discussion

Progress toward understanding basal forebrain contributions to brain function and behavior has benefited from approaches that support recordings from genetically identified cholinergic and GABAergic

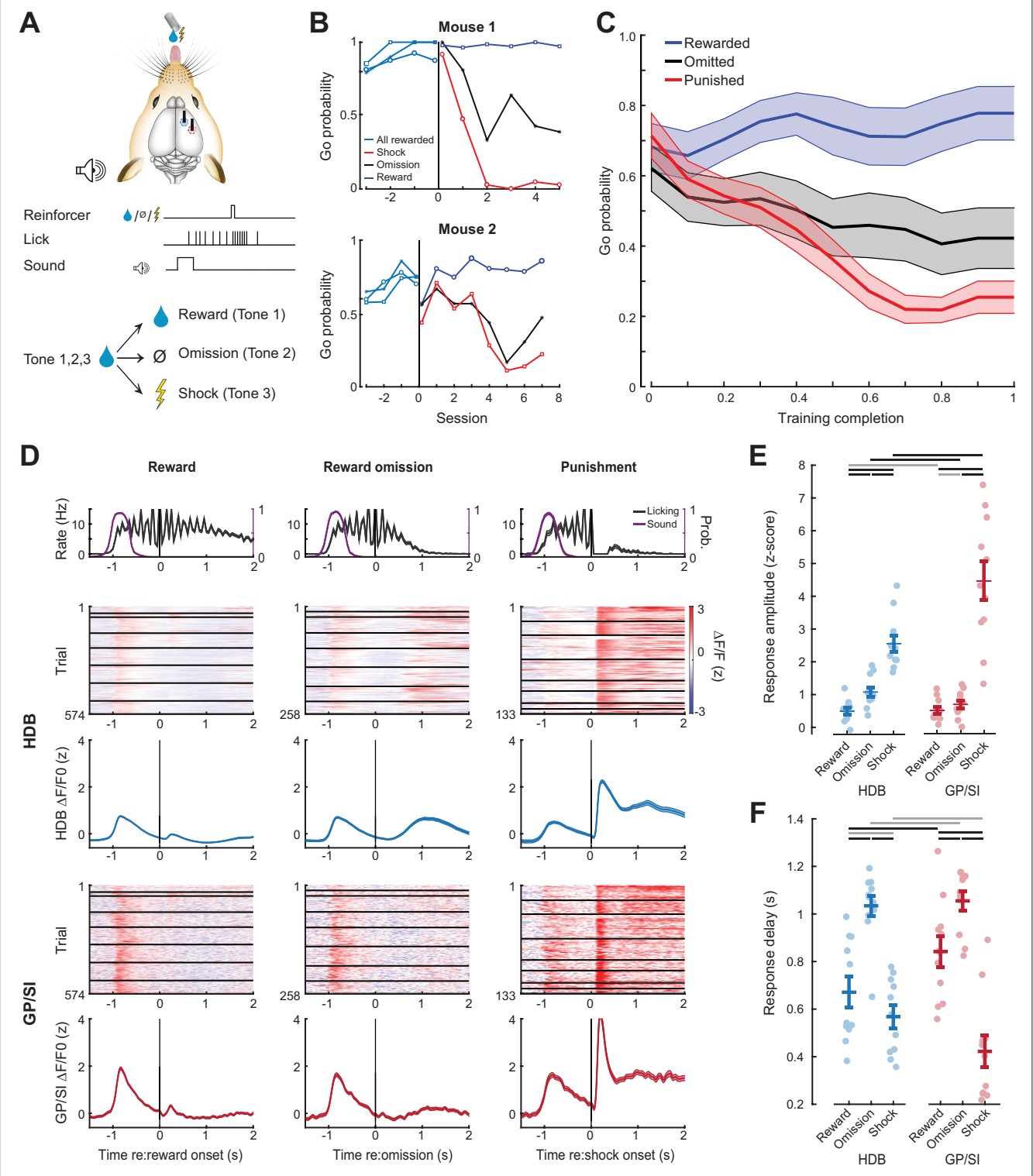

**Figure 6.** Differential responses of horizontal limb of the diagonal band of Broca (HDB) and globus pallidus and substantia innominata (GP/SI) basal forebrain cholinergic neurons (BFCNs) to reward, punishment, and reward omission. (**A**) Once mice were reliably licking for reward following the onset of the low-, mid-, or high-frequency tone, the reinforcement outcome was changed such that ≥7 licks in 2.8 s elicited a tongue shock for one frequency and the omission of reward for the other. (**B**) Go (≥7 licks in 2.8 s) probability for all three tones when they were all associated with reward and after the reinforcement outcome was changed for two of the tones. Data are shown for two mice that modify the behavior to the change in reinforcement outcome at different rates. Vertical line denotes the transition from all rewarded (Phase 1) to variable outcome (Phase 2). Circle, asterisk, and squares

*Figure 6 continued on next page*

*Figure 6 continued*

indicate low-, mid-, and high-frequency tones, respectively. (**C**) Mean ± SEM Go probability for each reinforcement outcome as fraction of training completed in N = 11 mice. (**D**) Tone-evoked cholinergic GCaMP responses from HDB (*rows 2–3*) and GP/SI (*rows 4–5*) of a single mouse from 965 Go trials distributed over eight behavioral sessions following the change in reinforcement outcome. All data are plotted relative to reinforcement onset. *Top row:* Timing of lickspout activity (black) and tone onset probability (purple). *Rows 2 and 4:* Heatmaps of single trial fractional change values in HDB (row 2) and GP/SI (row 4). Horizontal black lines in heatmaps denote different daily recording sessions. *Rows 3 and 5:* Mean ± SEM corresponding to each of the heatmaps above. Vertical lines denote reinforcement onset. (**E**) Reinforcement-related response amplitudes were calculated by subtracting the mean activity during a 2 s pre-stimulus baseline period from the peak activity occurring within 2 s following the 7th lick. Circles denote individual mice (N = 11 for all conditions), bars denote sample mean and SEM. Two-way repeated measures ANOVA with reinforcement type and structure as independent variables: Reinforcement type, F = 80.62, p = 3 × 10$^{-10}$; structure, F = 5.7, p = 0.03; reinforcement type × structure interaction, F = 8.01, p = 0.003. Black and gray horizontal lines denote significant (p < 0.05) and non-significant pairwise contrasts after correcting for multiple comparisons. (**F**) Reinforcement-related response latency was defined as the mean latency of the single trial peak responses relative to the offset of the 7th lick. Circles denote individual mice (N = 11 for all conditions), bars denote sample mean and SEM. Two-way repeated measures ANOVA with reinforcement type and structure as independent variables: Reinforcement type, F = 51.28, p = 1 × 10$^{-8}$; structure, F = 0.08, p = 0.78; reinforcement type × structure interaction, F = 7.52, p = 0.004. Black and gray horizontal lines denote significant (p < 0.05) and non-significant pairwise contrasts after correcting for multiple comparisons.

The online version of this article includes the following figure supplement(s) for figure 6:

**Figure supplement 1.** Basal forebrain responses on omission trials reflect reinforcement prediction error, not a motor-related signal.

cell types in behaving animals (*Yang et al., 2017*). Even when experiments are largely performed on a single species (mice) and focus largely on a single neurochemical cell type (cholinergic neurons), there have still been inconsistencies in the conclusions drawn from different experiments, particularly with respect to how BFCN activity relates to movement, to reward, to conditioned vs. unconditioned sensory cues, and to predicting behavioral outcomes from cue-related activity. We reasoned that this variability could reflect differences in measurement technique, inter-subject variation, and differences in where the recordings were made along the extent of the rostrocaudal basal forebrain. To address this possibility, we developed an approach to study all of the experimental features listed above in each of our subjects while making simultaneous recordings from rostral and caudal regions of the cholinergic basal forebrain that are known to have distinct afferent and efferent connections.

The findings reported here can be summarized by identifying experimental features where HDB was more strongly involved than GP/SI, where GP/SI was more strongly involved than HDB, and where both structures were equivalently responsive (*Figure 8*). HDB, perhaps on account of its strong reciprocal connectivity with the prefrontal cortex, showed a stronger involvement than GP/SI on variations of pupil-indexed internal brain state, in predicting whether the perceptual outcome in a behavioral detection task was a hit or a miss, and in encoding the omission of anticipated rewards (*Gielow and Zaborszky, 2017*; *Rye et al., 1984*; *Zaborszky et al., 2012*). Conversely, GP/SI, perhaps on account of stronger relative inputs from the striatum and thalamic regions encoding nociceptive inputs and auditory stimuli, showed a stronger functional selectivity for auditory stimuli, self-initiated movements, punishment, and learning-related plasticity of auditory cues associated with punishment (*Chavez and Zaborszky, 2017*; *Rye et al., 1984*; *Zaborszky et al., 2012*).

## Specialized processing in the caudal tail of the cholinergic basal forebrain

Among these statistically significant regional differences, many were differences of degree, but a few were more akin to differences of kind. In particular, 'native' BFCN responses to unconditioned auditory stimuli were markedly stronger in GP/SI compared with HDB, as was learning-related enhancement of punishment-predicting auditory cues. Other reports of BFCNs have either observed that cue-evoked responses only emerge after a learned association with reward (*Crouse et al., 2020*; *Kuchibhotla et al., 2017*; *Parikh et al., 2007*; *Sturgill et al., 2020*) or were not obviously present either for reward- or punishment-predicting cues (*Hangya et al., 2015*). Although anterior BFCNs receive sparse monosynaptic thalamic inputs from the medial subdivision of the medial geniculate body and neighboring posterior intralaminar nucleus (*Gielow and Zaborszky, 2017*), the input from auditory thalamic regions to GP/SI appears far more dense (*Chavez and Zaborszky, 2017*). Single unit recordings from GP/SI have revealed well-tuned short-latency (~10 ms) spiking responses to a broad class of sounds including moderate intensity tones and noise bursts (*Chernyshev and Weinberger, 1998*; *Guo et al., 2019*; *Maho et al., 1995*). By contrast, single unit recordings from basal

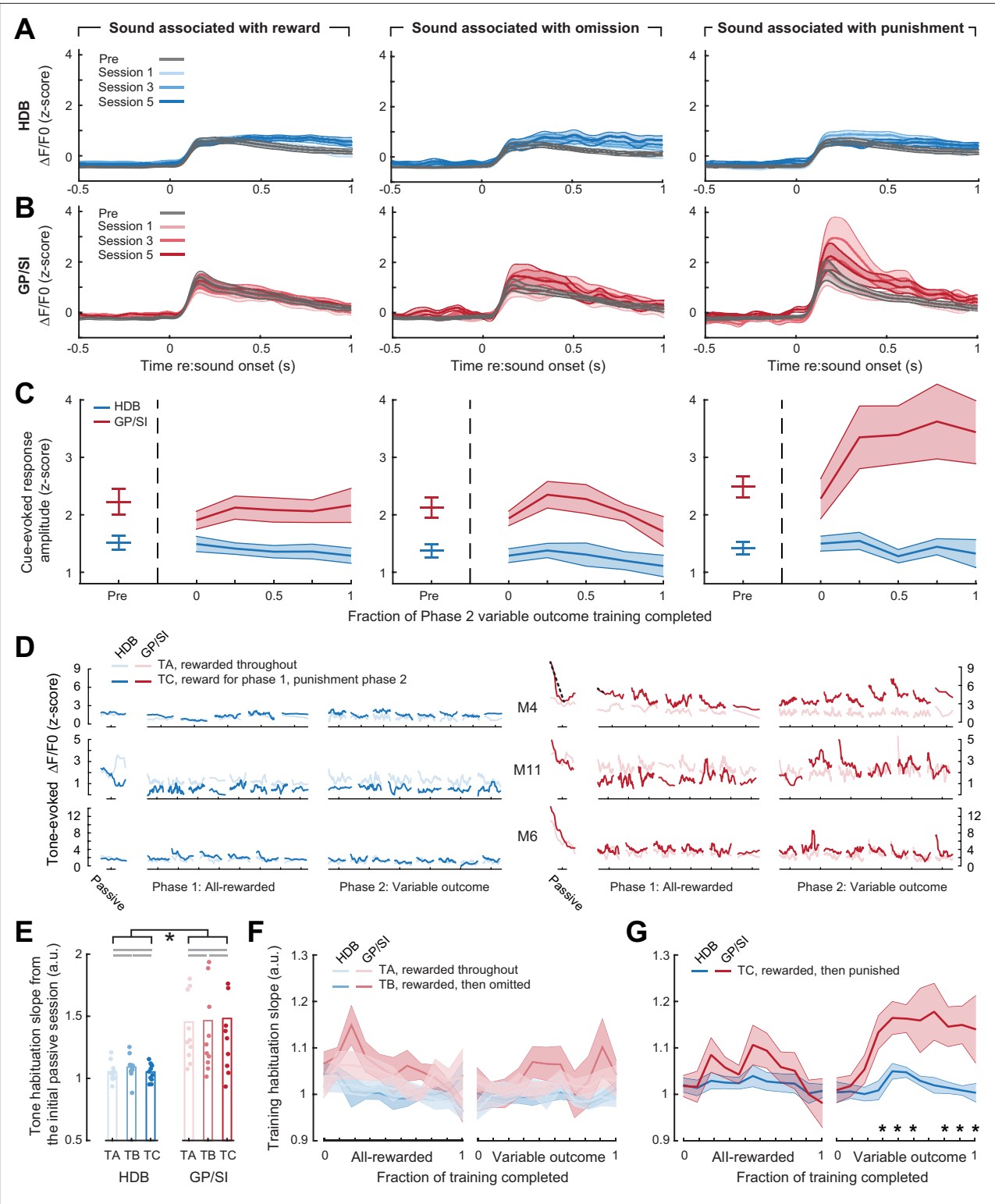

**Figure 7.** Enhanced basal forebrain cholinergic neuron (BFCN) responses to punishment-predicting cues in globus pallidus and substantia innominata (GP/SI), not horizontal limb of the diagonal band of Broca (HDB). (**A–B**) Mean ± SEM tone-evoked GCaMP activity in HDB (N = 11, **A**) and GP/SI (**B**) for the tone frequency associated with reward (left column), reward omission (middle column), and punishment (right column). Mean cue-evoked responses are shown during Phase 1 of the task, in which all frequencies were associated with reward (gray), and for three subsequent sessions following the transition to Phase 2, where variable reinforcement outcomes were introduced. (**C**) Mean ± SEM tone-evoked response amplitudes in HDB and GP/SI (N = 11) were calculated by subtracting the mean activity during a 2 s pre-stimulus baseline period from the peak of activity within 400 ms of

*Figure 7 continued on next page*

*Figure 7 continued*

sound onset. Phase 2 behavior sessions were assigned to one of five different discrete time bins according to the fraction of total training completed. Learning-related enhancement was only noted for the punishment-predicting tone in GP/SI (three-way repeated measures ANOVA with training time, reinforcement type, and structure as independent variables: main effect for training time, F = 1.62, p = 0.18; main effect for reinforcement type, F = 3.99, p = 0.03; main effect for structure, F = 23.38, p = 0.0006; training time × reinforcement type × structure interaction, F = 2.2, p = 0.04). (**D**) Within- and between-session dynamics in tone-evoked HDB (*left*) and GP/SI (*right*) responses are shown during the initial passive characterization session (see *Figure 2*) and all subsequent Phase 1 and Phase 2 training sessions for three mice exemplifying varying degrees of enhanced GP/SI response amplitude and habituation for punishment-predicting sounds. Mouse (M) number corresponds to the fiber locations shown in *Figure 1—figure supplement 1*. Each individual line presents the smoothed average (7-point median filter) for all trials within a given behavioral session for two tone frequencies. Dashed lines denote the linear slope measurement for within-session habituation. (**E**) Within-session habituation of tone-evoked responses during the initial passive characterization session, measured as the linear slope over the first 10 trials. Tones (T) A, B, and C denote the frequencies that will ultimately be associated with reward, omission, and punishment in Phase 2 of the operant task. Habituation is significantly greater in GP/SI than HDB but does not differ between tone frequencies (two-way repeated measures ANOVA with structure and tone frequency as independent variables: main effect for structure, F = 13.41, p = 0.004 [denoted by black lines and asterisk]; main effect for tone frequency, F = 0.08, p = 0.92). Gray horizontal lines denote non-significant pairwise differences after correcting for multiple comparisons (p > 0.58 for each). (**F**) Within-session habituation of tone-evoked responses during Phases 1 and 2 of the operant task, measured as the linear slope from the first 20% of trials within each session. Mean ± SEM habituation slope for frequencies associated with reward and reward omission are not changed over time or reinforcement type (three-way ANOVA with time, reinforcement type, and structure as independent variables: main effect for reinforcement type, F = 1.0, p = 0.34; main effect for time; F = 0.77, p = 0.66; N = 11).

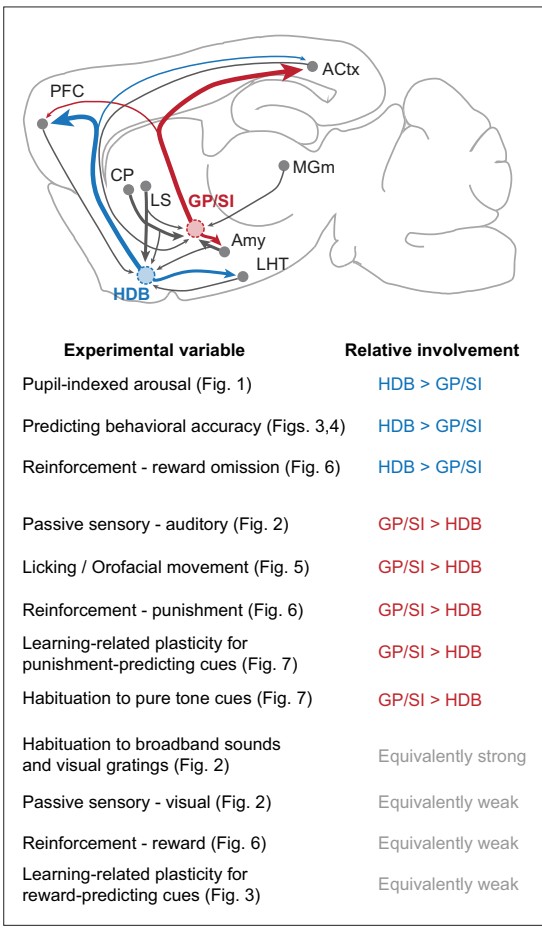

**Figure 8.** Summary of functional specializations in the rostral and caudal basal forebrain. A summary of the relative involvement of horizontal limb of the diagonal band of Broca (HDB) and globus pallidus and substantia innominata (GP/SI) across all experimental variables tested in this study.

forebrain units in the medial septum (rostral to HDB) also identified short-latency responses to unconditioned sounds. However, medial septal units only responded to intense broadband sounds and were derived from pontine central gray afferent inputs, not auditory thalamic regions (*Zhang et al., 2018*).

Given that GP/SI is the predominant source of BFCN input to lateral neocortical regions including ACtx, one clear implication that would be important to test in future studies is that auditory stimuli – even sounds with no behavioral relevance – should elicit ACh release in ACtx. ACh acts through local ACtx microcircuits to remove the fetters that normally limit long-term associative plasticity, thereby enabling local synaptic processes that support auditory fear memory encoding (*Letzkus et al., 2015*; *Weinberger, 2004*) and perceptual learning (*Froemke et al., 2013*; *Takesian et al., 2018*). Importantly, learning-related plasticity in ACtx requires transient neuromodulatory surges and does not occur when stimuli are presented in a passive context (*Froemke, 2015*). This suggests that cholinergic regulation of cortical plasticity is not an all-or-none gating process but instead may reflect a threshold that is only exceeded when sound-evoked cholinergic inputs are themselves transiently amplified through learning (*Figure 7C and G*, *Guo et al., 2019*). Beyond simple gating mechanisms, dual recordings from single BFCNs and ACtx neurons during Pavlovian auditory learning (*Guo et al., 2019*) and attentionally demanding auditory tasks (*Laszlovszky et al., 2020*) have demonstrated dynamics in BFCN-cortex synchrony that change in lockstep with associative plasticity and auditory perceptual salience. Although the upstream factors that regulate BFCN plasticity and inter-regional synchrony have yet to be identified, it is clear that models portraying phasic cortical ACh release occurring only at times of reward, punishment, or heightened arousal need to be reevaluated, at least as they relate to the caudal tail of the basal forebrain and the ACtx.

As for the learning-related enhancement of punishment-predicting – but not reward-predicting cues in GP/SI – this again may reflect the unique input this region of the basal forebrain receives from the medial geniculate and intralaminar thalamic groups, which also exhibit rapid, selective, and long-lasting enhanced spiking to tones associated with aversive stimuli (*Edeline and Weinberger, 1992*; *Weinberger, 2011*). Learned enhancement of reward-predicting auditory cues have either been observed in BFCN axons arising from more rostral basal forebrain regions that innervate the basolateral amygdala (*Crouse et al., 2020*) and ACtx (*Kuchibhotla et al., 2017*), or have only been described in putative non-cholinergic neurons in HDB (*Lin and Nicolelis, 2008*), which therefore offers no point of contradiction with the absence of reward-related enhancement reported here in HDB and GP/SI BFCNs. As mentioned above, another possibility is that HDB and GP/SI BFCNs did exhibit an increased response to reward-predicting cues during the initial association of sound and reward, which occurred during the behavioral shaping period when we did not monitor activity. Collectively, these findings point toward the caudal tail of the basal forebrain, which provides the strongest overall projection from the basal forebrain to ACtx and where approximately 80% of the neurons are cholinergic (*Guo et al., 2019*; *Kamke et al., 2005*; *Rye et al., 1984*), as a hub for encoding and associating sound with aversive, noxious stimuli, and for regulating inhibitory microcircuits within ACtx for long-term plasticity to enhance the representation of threat-predicting sounds (*David et al., 2012*; *Guo et al., 2019*; *Letzkus et al., 2011*).

Collectively, our findings support the view that rostral and caudal BFCN responses share many similarities in their response features, particularly as they relate to arousal and reinforcement, yet regional afferent and efferent connectivity differences – particularly in the caudal tail of the basal forebrain – support regional specializations for encoding sensory salience and expressing associative plasticity during aversive reinforcement learning. Interestingly, a neighboring region to GP/SI in the tail of the striatum also receive specialized dopaminergic inputs that do not encode reward value, but rather are activated by potentially threatening sensory stimuli (*Menegas et al., 2018*). This raises the interesting suggestion that cholinergic and dopaminergic signaling in the caudal tail of the rodent basal ganglia and basal forebrain may function as a hub for encoding threatening signals and selecting adaptive threat avoidance behaviors (*Watabe-Uchida and Uchida, 2018*).

## Technical considerations in the interpretation of these findings

From a technical perspective, fiber-based imaging was the best methodology to address our experimental aims, particularly for the goal of performing simultaneous measurements of rostral and caudal BFCNs over an extended period. BFCNs in GP/SI are arrayed in a thin dorsoventral sheet along the lateral wall of the internal capsule and then split into thin vertically oriented arrangements along

the medial and lateral boundaries of the external GP (*Clayton et al., 2021*; *Guo et al., 2019*). This anatomy is not optimal for endoscopic imaging through implanted lenses, as it could be challenging to visualize BFCNs in a single focal plane. Two-photon imaging of the cortical axon terminals from GP/SI BFCNs is feasible (*Nelson and Mooney, 2016*), though these signals would still arise from an indeterminate number of neurons and concerns about tissue bleaching and photodamage would not be compatible with the hours of daily testing over 30+ consecutive days that was performed here. Antidromic or somatic optogenetic tagging of single BFCNs is the gold standard, affording the highest level of spatial and temporal resolution. Our prior work used the antidromic variant of this approach to make targeted single unit recordings from GP/SI BFCNs that project to ACtx, but the yield was punishingly low (~1% of all units recorded) and units could not be held long enough to measure responses to all of the experimental variables tested here (*Guo et al., 2019*).

However, there are important limitations and technical caveats with fiber-based bulk GCaMP imaging that should be taken into consideration in the interpretation of these findings. Because fiber photometry signals arise from populations of neurons, it is impossible to discern whether differences in response amplitude over learning or across different behavioral states reflect the activation of privileged ensembles that were hitherto silent or instead an increased response expressed uniformly across neurons. Conversely, the absence of differences in the population signal could belie striking shifts in the representational dominance of antagonistically related cellular ensembles that would not be captured by changes in net signal amplitude (*Grewe et al., 2017*; *Gründemann, 2021*; *Taylor et al., 2021*). Another caveat in the interpretation of fiber-based GCaMP imaging is that the slow temporal kinetics and poor spatial resolution combines somatic and neuropil-based calcium signals and obscures the relationship to spike rates in distinct types of BFCNs. This would be particularly worrisome if the axons of BFCNs in HDB or GP/SI projected to or through the other region, as this could either produce optical cross-talk (i.e., axon fluorescence originating from BFCNs in region A measured on the region B fiber) or functional cross-talk (i.e., projections from BFCNs in region A modulate the activity of region B). Neither of these possibilities is likely a concern in the interpretation of these findings. Correlating all single trial tone-evoked response amplitudes measured on each fiber reveals a very weak association ($R^2$ = 0.16, n = 21,099 trials), demonstrating that the activity in HDB and GP/SI can be measured independently. Further, anatomical characterizations suggest that BFCN inputs within the basal forebrain primarily arise from local neurons rather than remote regions (*Gielow and Zaborszky, 2017*). To this point, direct visualization of efferent HDB axons showed that they left the basal forebrain in a medial and dorsal orientation, coming nowhere near the GP/SI fiber (*Bloem et al., 2014*).

## Cholinergic regulation of perceptual salience

Although the proportion of cholinergic neurons declines rostral to GP/SI, the overall spatial arrangement and larger cell body size of BFCNs in nucleus basalis and HDB makes somatic optogenetic tagging of single units somewhat more feasible (*Hangya et al., 2015*; *Laszlovszky et al., 2020*). An elegant recent study has demonstrated that BFCNs within nucleus basalis and HDB are not an indivisible class, but can themselves be further subdivided into bursting and regular-firing BFCNs, where the proportion of each type varied across the rostral-caudal extent of the basal forebrain and had distinct patterns of synchronization both with respect to each other and with network oscillations measured in ACtx (*Laszlovszky et al., 2020*). Interestingly, when studied in the context of an auditory task similar to the paradigm used here, the spike timing of bursting BFCNs showed a stronger coupling with the ACtx on trials where mice made a Go response (regardless of whether it was a hit or false positive) whereas the regular-firing BFCNs showed a stronger coupling with the ACtx on trials where mice made the correct response (regardless of whether it was Go or NoGo).

Although fiber-based BFCN imaging cannot distinguish between the involvement of each cell type, we also noted a striking correspondence between GCaMP activity in the peri-cue period and the subsequent behavioral outcome (either hit or miss, *Figure 3H–I*). Our findings confirm an association between BFCN activity and trial outcome in the period following the delivery of the auditory cue (*Laszlovszky et al., 2020*), but we observed a clear connection to trial outcome during the preceding 1 s baseline period (thereby obviating any confound related to differences in licking activity between hit and miss trials). Prior studies have also reported that cholinergic levels prior to auditory onset can predict whether the animal would subsequently produce the correct or incorrect operant response,

suggesting the bulk measures may be sensitive to pre-cue dynamics that are not resolvable at the level of single neurons (**Kuchibhotla et al., 2017**; **Parikh et al., 2007**). In a recent study, we found that hit or miss trial outcomes in a challenging auditory detection task could be predicted from the degree of synchrony in local networks of ACtx layer 2/3 pyramidal neurons measured from a 1 s period prior to the delivery of the auditory cue (**Resnik and Polley, 2021**). As the cholinergic basal forebrain has classically been studied as a master regulator of cortical network synchrony (**Buzsaki et al., 1988**; **Metherate et al., 1992**), one clear suggestion is that ongoing cholinergic dynamics in the period preceding environmental sensory cues strongly regulate cortical network state, which can have profound impacts on the accurate encoding of sensory cues and appropriate selection of cue-directed actions.

# Materials and methods

## Key resources table

| Reagent type (species) or resource | Designation | Source or reference | Identifiers | Additional information |
|---|---|---|---|---|
| Genetic reagent (*Mus musculus*) | B6.129S-*Chat*^tm1(cre)Lowl/MwarJ | Jackson Laboratory | RRID:IMSR_JAX:031661 | Male |
| Genetic reagent (*Mus musculus*) | B6.Cg-Igs7^tm148.1(tetO-GCaMP6f,CAG-tTA2)Hze/J | Jackson Laboratory | RRID:IMSR_JAX:030328 | Female |
| Antibody | Anti-ChAT (goat polyclonal) | Millipore Sigma | Cat #: AB144P RRID: AB_2079751 | (1:100) |
| Antibody | Anti-Goat (donkey polyclonal) | Abcam | Cat#: AB150132 RRID: AB_2810222 | (1:500) |
| Recombinant DNA reagent | ACh sensor | Dr Yulong Li | GRAB$_{ACh}$3.0 | |
| Software, algorithm | Labview | National Instruments | RRID: SCR_014325 | Version 2015 |
| Software, algorithm | MATLAB | Mathworks | RRID: SCR_001622 | Version R2021a |
| Other | DAPI stain | Vectorlabs | Cat #: H-1500–10 RRID:AB_2336788 | |
| Other | Allen Brain Atlas | **Lein et al., 2007** | RRID:SCR_013286 | |

## Animals

All procedures were approved by the Massachusetts Eye and Ear Animal Care and Use Committee and followed the guidelines established by the National Institutes of Health for the care and use of laboratory animals. Male ChAT-cre-ΔNeo (homozygous, Jackson Labs 031661) and female Ai148 mice (hemizygous, Jackson Labs 030328) were bred in-house to generate mice of both sexes for this study. Offspring were therefore hemizygous for ChAT-cre-ΔNeo and either had hemizygous expression of cre-dependent GCaMP6f (ChAT+/GCaMP+) or did not express GCaMP (ChAT+/GCaMP-). Offspring genotypes were confirmed by PCR (Transnetyx probes) and by histology following perfusion.

Experiments were performed in adult mice, 2–3 months of age at the time the first measurement was performed. Prior to behavioral testing, mice were maintained on a 12 hr light/12 hr dark cycle with ad libitum access to food and water. Mice were grouped-housed unless they had undergone a major survival surgery. Dual fiber imaging of ChAT neuron GCaMP fluorescence in GP/SI and HDB was performed in 11 ChAT+/GCaMP+ mice, four of which were used for additional histological quantification. Fiber imaging of ACh3.0 sensor fluorescence in ACtx was performed in 10 ChAT+/GCaMP- mice.

## Surgical procedure for GCaMP photometry

Mice were anesthetized with isoflurane in oxygen (5% induction, 2% maintenance) and placed in a stereotaxic frame (Kopf Model 1900). A homeothermic blanket system was used to maintain body temperature at 36.6° (FHC). Lidocaine hydrochloride was administered subcutaneously to numb the scalp. The dorsal surface of the scalp was retracted and the periosteum was removed. Dual optic fiber implants (Doric, 400 µm core, NA 0.48, 1.25 mm diameter low-autofluorescence metal ferrule) were slowly lowered into HDB (0.9 × 0.3 × 4.7) and GP/SI (2.5 × –1.5 × 3.3 mm from bregma, [lateral × caudal × ventral]) in the right hemisphere. Silicon adhesive (WPI Kwik-Sil) was applied to the exposed brain surface. The exposed skull surface was prepped with etchant (C&B metabond) and 70% ethanol

before affixing a titanium head plate (iMaterialise) to the skull with dental cement (C&B Metabond). At the conclusion of the procedure, Buprenex (0.05 mg/kg) and meloxicam (0.1 mg/kg) were administered and the animal was transferred to a warmed recovery chamber.

## Surgical procedure for ACh sensor photometry

The initial surgical procedures and perioperative care were similar to that for GCaMP photometry. The skull overlying the right ACtx exposed by moving the temporalis muscle. A burr hole was made on the temporal ridge at 2.9 mm posterior to bregma, using a 31-gauge needle. A motorized injection system (Stoelting) was used to inject 200 nL of AAV9-hSyn-ACh3.0 (diluted 10% in sterile saline from $3.45 \times 10^{13}$ genome copies/mL) via a pulled glass micropipette 0.5 mm below the pial surface. We waited at least 10 min following the injection before withdrawing the micropipette. A tapered fiber (Optogenix, NA 0.39, diameter 200 µm, active length 1.0 mm) was implanted 1 mm below the pial surface and secured using dental cement dyed with India Ink, which also secured the titanium head plate. Sensor photometry experiments began 3 weeks following the injection.

## Pupillometry

Mice were placed in an electrically conductive cradle and habituated to head fixation during three sessions of 15, 30, and 60 min over 3 consecutive days. Video recordings of the pupil under isoluminous background conditions were performed during the final habituation session and the following sensory characterization day. Video recordings were made at 30 Hz with a CMOS camera (Teledyne Dalsa, model M2020) outfitted with a lens (Tamron 032938) and infrared longpass filter (Midopt lp830-25.5). Automated analysis of pupil diameter follows the procedure described previously by *McGinley et al., 2015a*. Briefly, each movie was thresholded such that most pixel values within the pupil were below threshold and all other pixels were above threshold. A circle was fit to the pupil by first calculating the center of mass within the pupil and then centering a circle with the corresponding area to that point. Canny edge detection was then used to identify edge pixels within each grayscale image. Edge pixels were removed if they were more than three pixels away from a pupil pixel or outside of an annulus with diameters that were 0.5 and 1.75 the diameter of the initial fit circle. As a final step, an ellipse was fit to the remaining edge pixels using least-squares regression and the pupil diameter was defined from the diameter of a circle with a matching area. This procedure was performed for each image frame using a MATLAB (Mathworks) script adapted from the original publication (*McGinley et al., 2015a*).

Pupil diameter for ACh3.0 sensor imaging experiments was extracted using DeepLabCut (version 2.1.8.2, *Nath et al., 2019*). Specifically, three investigators each labeled 100 frames taken from 10 mice, for a total of 300 frames from 30 mice. The four cardinal and four intercardinal compass points were marked for each pupil. Marker placement was confirmed by at least one additional investigator. Training was performed on 95% of frames. We used a ResNet-101 based neural network with default parameters for 1,030,000 training iterations. We then used a p-value cutoff of 0.9 to condition the X,Y coordinates for analysis. This network was then used to analyze videos from similar experimental settings from the 10 ACh3.0 sensor imaging mice. We calculated pupil diameter for each frame by fitting an ellipse to the identified pupil contour points using a least-squares criterion and calculating the long axis diameter.

## Operant behavioral testing

All mice proceeded through the same series of tests beginning 2 weeks following fiber implant surgery (*Figure 2A*). On sessions 1 and 2, mice were habituated to head fixation and the body cradle. On session 3, pupillometry was performed without sensory stimulation. On session 4, pupillometry and fiber imaging was performed in response to the presentation of auditory or visual stimuli. Beginning on day 5, mice were placed on water restriction and were monitored until they reached 80% of their baseline weight. Beginning on day 8 or 9, after several days of behavioral shaping, mice began appetitive operant training that rewarded vigorous licking shortly following the presentation of three different tone frequencies. Finally, on days 13–22, mice were switched to a reinforcement reversal task, where two of the previously rewarded frequencies were switched to neutral or aversive reinforcement. These methods for each of these stages are provided in detail below.

## Sensory characterization

Visual gratings were generated in MATLAB using the Psychtoolbox extension and presented via an 800 × 480 pixel display (Adafruit 2406) positioned approximately 15 cm from the left eye 45° off midline. Visual gratings were presented with a spatial frequency of 0.035 cycles per degree at three contrasts: 11%, 33%, and 100%. Gratings (2 s duration) were presented at both vertical and horizontal orientations. Spatial drift (2 Hz) was imposed along the orthogonal axis to orientation.

Auditory stimuli were either pure tones or auditory drifting gratings (i.e., ripples). Stimuli were generated with a 24-bit digital-to-analog converter (National Instruments model PXI-4461) and presented via a free-field speaker (CUI, CMS0201KLX) placed approximately 10 cm from the left (contralateral) ear canal. Free-field stimuli were calibrated using a wide-band free-field microphone (PCB Electronics, 378C01). Pure tones were low (either 6 or 6.8 kHz), mid (9.5 or 11.3 kHz), or high (13.9 or 18.5 kHz) frequencies presented at three intensities (30, 50, and 70 dB SPL). Tones were 0.4 s duration shaped with 5 ms raised cosine onset and offset ramps. Auditory gratings ranged from 2 to 45 kHz with 2 s duration (5 ms raised cosine onset and offset ramps), presented at downward and upward frequency trajectories (at –2 and +2 Hz) at three intensities (30, 50, and 70 dB SPL). The spectrum was shaped with 20 frequency carriers per octave that were sinusoidally modulated with 90% depth at one cycle per octave.

A single block consisted of 22 unique stimulus trials with a 7 s inter-trial interval (6 visual gratings [2 orientations × 3 contrasts], 9 tones [3 frequencies × 3 levels], 6 auditory gratings [2 directions × 3 intensities] and one silent trial where neither an auditory nor visual stimulus was presented). The stimulus order was randomly determined for each of 20 presentation blocks.

## Operant training

Behavioral shaping for the rewarded tone detection task began after the sensory characterization session. In the initial phase of training, mice learned to vigorously lick a spout shortly following tone onset (low, mid, or high frequency, as specified above at 70 dB SPL) in order to receive a liquid reward (10% sucrose in water, 1.5 μL per reward, 1 reward per trial). Initially, tones were paired with rewards (i.e., Pavlovian conditioning), initiated 0.5 s after tone onset. Fiber imaging was not performed during behavioral shaping.

Once mice were reliably licking prior to reward onset, the requirement to trigger reward delivery (i.e., operant conditioning) was progressively increased. The licking criterion to receive a reward was seven lickspout contacts within a 2.8 s period beginning 0.2 s after stimulus onset, where the interval between any 2 consecutive licks could not exceed 1 s. Individual trials were scored as hits, according to the criterion above, misses (no licks), or partial hits (lickspout contact that did not meet the criterion above). Intertrial intervals were determined randomly from a truncated exponential distribution within a range of 7–10 s. Trials were aborted in the event of lick spout contact in a withhold period of 2 s (initial phase) or 1.5 s (after reversal) preceding stimulus onset. Generally, mice learned to produce 7 licks in 2.8 s to initiate reward with low false alarm rates within two to three sessions.

In order to analyze licking-related activity, separate lick bouts were also selected from the intertrial periods. Lick bouts were defined as at least 2 lick contacts less than 250 ms apart, bookended by quiescent lick-free periods at least 1 s each before and after the bout.

Once the reward rate exceeded 70% across all frequencies for at least one session, mice were transitioned to the reversal stage of the operant task in which one of the three tones remained associated with reward, one was associated with shock, and a third was not associated with reward or punishment (i.e., neutral outcome). The assignment of tone frequency to reinforcement condition was randomized across mice. Punishment was delivered by briefly electrifying the lick spout (0.6mA for 0.4 s) once the lick bout threshold (7 licks in 2.8 s) had been crossed. During this stage, the rewarded tone was presented on 50% of trials and the punished and neutral tones were each presented on 25% of trials. Operant testing was terminated once the Go probability stabilized across all tone frequencies for at least 2 consecutive days.

## Fiber photometry

### Data acquisition

LEDs of different wavelengths provided a basis for separating calcium-dependent (465 nm) and calcium-independent (405 nm) fluorescence. Blue and purple LEDs were modulated at 210 and

330 Hz, respectively, and combined through an integrated fluorescence mini-cube (FMC4, Doric). The power at the tip of the patch cable was 0.1–0.2 mW. The optical patch cable was connected to the fiber implant via a zirconia mating sleeve. Bulk fluorescent signals were acquired with a femtowatt photoreceiver (2151, Newport) and digital signal processor (Tucker-Davis Technologies RZ5D). The signal was demodulated by the lock-in amplifier implemented in the processor, sampled at 1017 Hz, and low-pass filtered with a corner frequency at 20 Hz. The optical fibers were prebleached overnight by setting both LEDs to constant illumination at a low power (<50 uW).

## Data processing

After demodulation, the 465 nm GCaMP responses were calculated as the fractional change in fluorescence DF/$F_0$, where $F_0$ was defined as the running median fluorescence value in a 60 s time window. DF/$F_0$ traces were then low-pass filtered with a second-order zero-lag Butterworth filter, with a cutoff frequency set to 7 Hz. Event-related DF/$F_0$ values were then z-scored relative to baseline activity levels. For passive sensory characterization experiments, the baseline distribution consisted of all DF/$F_0$ recorded during a 2 s window preceding visual or auditory stimulus onset. For recordings made during the operant task, the baseline distribution consisted of all DF/$F_0$ recorded during a 2 s period prior to auditory cue onset that was combined across trial types and sessions.

## Data analysis

To measure the relationship with spontaneous pupil fluctuations (*Figure 1*), photometry data were first downsampled to 30 Hz before measuring coherence with a hamming window of 1500 samples and 1400 sample overlap. Lag was defined by the peak of the cross-correlation between fluorescence (GCaMP or ACh3.0) and pupil fluctuations. Event-related response amplitudes (*Figures 2–7*) were calculated on an individual trial basis. To measure sensory-evoked response amplitudes (*Figure 2C* and *Figure 2—figure supplement 1*), the mean fractional change during a 2 s pre-stimulus baseline was subtracted from both the peak fractional change during the 2 s stimulus period and a 0.4 s period immediately preceding stimulus onset. The sensory-evoked response amplitude was then calculated as post−pre. The amplitude of spontaneous transients (*Figure 2G*) was calculated on trials where neither an auditory nor visual stimulus was presented. A threshold was applied to DF/$F_0$ values for each trial to identify time points corresponding to the bottom 5% of fractional change values. Spontaneous transients were operationally defined as any time point containing a value that was at least 0.5 z-scores above the 5% threshold. Spontaneous activity was then quantified as the mean value for all suprathreshold values within the trial. Time windows used to calculate the various event-related response amplitudes related to the behavioral task (*Figures 3–7*) are defined in the corresponding figure legends.

To determine whether BFCN activity in the period just before or just after presentation of the target sound could be used to classify behavioral outcomes (hit vs. miss), we used an SVM classifier with a linear kernel. We fit the classifier model to a data matrix consisting of the fractional change in fluorescence (binned at 10 ms resolution) either during a 1 s period preceding tone presentation or a 400 ms period following the onset of tone presentation. We used principal components analysis to reduce dimensionality of the data matrix before classification. We then used only the principal components needed to account for 90% of the variance in the data for the SVM-based classification. Leave one-out cross-validation was then used to train the classifier and compute a misclassification rate on the untrained trial. This process was then iterated 50 times, each time ensuring an equivalent number of hit and miss trials in the sample by randomly downsampling the hit trials. We repeated this process for each imaging session independently and calculated the mean decoding accuracy across sessions for each mouse. As a control we randomly assigned the hit and miss labels to confirm that classification accuracy was at chance. The SVM training and cross-validation procedure was carried out in MATLAB using the 'fitcsvm' and 'predict' functions.

## Histology

At the conclusion of imaging, mice were deeply anesthetized and prepared for transcardial perfusion with a 4% formalin solution in 0.1 M phosphate buffer. The brains were extracted and post-fixed at room temperature for an additional 12 hr before transfer to 30% sucrose solution. For all brains, the

location of the fiber tip center was identified and translated to a reference atlas of the adult mouse brain created by the Allen Institute for Brain Science (as shown in *Figure 1—figure supplement 1*).

In a subset of brains (N = 4), coronal sections (30 μm) were rinsed for 1 hr in 0.1 M phosphate buffered saline (PBS) and 0.4% Triton-X, and then permeabilized for 1 hr with 1% Triton-X and 5% normal horse serum. Sections where incubated overnight in blocking solution containing the primary antibodies (Goat anti-ChAT 1:100, Millipore, AB144P). Sections were rinsed in PBS then incubated for 2 hr at room temperature in blocking solution containing secondary antibodies, counterstained in DAPI for 5 min, rinsed in PBS, mounted onto glass slides, and then coverslipped. Co-localization of ChAT and GCaMP was quantified in the HDB and GP/SI regions of interest immediately beneath the fiber tip and in the corresponding region in the contralateral hemisphere. Quantification of ChAT and GCaMP was also performed in the striatum from both hemispheres of the same sections. Regions of interest were imaged at 63× using a Leica DM5500B fluorescent microscope. Tiled image stacks were then separated into individual fluorophore channels and labeled cells were manually counted in each channel independently using Adobe Photoshop.

## Statistics

All statistical analyses were performed in MATLAB 2016b (Mathworks). Data are reported as mean ± SEM unless otherwise indicated. Inflated familywise error rates from multiple comparisons of the same sample were adjusted with the Holm-Bonferroni correction. Statistical significance was defined as $p < 0.05$. For fiber-based imaging, we did not exclude any trials or mice from our analysis. For pupil imaging during BFCN calcium recordings, four mice were excluded from pupil analysis because the automated algorithm failed to identify the perimeter of the pupil.

## Acknowledgements

These studies were supported by NIH grant DC017078 (DP), The Nancy Lurie Marks Family Foundation (DP), a Herchel Smith Harvard Scholarship (BR), a Fondation Zdenek et Michaela Bakala Scholarship (BR), and NIH grant K08MH116135 (EK).

BR collected and analyzed the combined calcium imaging, pupillometry and behavioral data. EK, YW, and TC collected and analyzed the combined ACh3.0 sensor and pupillometry data. MJ and YL developed the ACh3.0 sensor purchased for use in these experiments. BR and DP designed the experiments. DP and BR prepared the figures. DP wrote the manuscript, with input from all authors.

We thank Ken Hancock for programming additional changes in his behavioral neurophysiology data collection software. We thank Matt McGinley for hardware advice and software support for pupil diameter quantification. We thank Troy Hackett for support developing our immunolabeling and histology quantification protocols.

## Additional information

### Funding

| Funder | Grant reference number | Author |
| --- | --- | --- |
| National Institute on Deafness and Other Communication Disorders | DC017078 | Daniel B Polley |
| Nancy Lurie Marks Family Foundation | | Daniel B Polley |
| Herchel Smith Harvard Scholarship | | Blaise Robert |
| Fondation Zdenek et Michaela Bakala Scholarship | | Blaise Robert |
| National Institute of Mental Health | K08MH116135 | Eyal Y Kimchi |

| Funder | Grant reference number | Author |
|--------|------------------------|--------|

The funders had no role in study design, data collection and interpretation, or the decision to submit the work for publication.

## Author contributions
Blaise Robert, Data curation, Formal analysis, Investigation, Software, Visualization, Writing - review and editing; Eyal Y Kimchi, Data curation, Formal analysis, Investigation, Software, Writing - review and editing; Yurika Watanabe, Tatenda Chakoma, Investigation, Software, Writing - review and editing; Miao Jing, Yulong Li, Resources, Writing - review and editing; Daniel B Polley, Conceptualization, Funding acquisition, Project administration, Resources, Supervision, Visualization, Writing - original draft, Writing - review and editing

## Author ORCIDs
Blaise Robert (ID) http://orcid.org/0000-0001-7945-8775
Daniel B Polley (ID) http://orcid.org/0000-0002-5120-2409

## Ethics
All procedures were approved by the Massachusetts Eye and Ear Animal Care and Use Committee (protocol #10-03-006A) and followed the guidelines established by the National Institutes of Health for the care and use of laboratory animals.

## Decision letter and Author response
Decision letter https://doi.org/10.7554/eLife.69514.sa1
Author response https://doi.org/10.7554/eLife.69514.sa2

# Additional files

## Supplementary files
• Transparent reporting form

## Data availability
Figure 1—source data 1 contains the data for Figure 1D. All data generated or analyzed during this study are available on Mendeley Data (doi:https://doi.org/10.17632/d8tjdxyjcm.2).

The following dataset was generated:

| Author(s) | Year | Dataset title | Dataset URL | Database and Identifier |
|-----------|------|---------------|-------------|-------------------------|
| Robert B, Kimchi EY, Watanabe Y, Chakoma T, Jing M, Li Y, Polley DB | 2021 | A functional topography within the cholinergic basal forebrain for encoding sensory cues and behavioral reinforcement outcomes | https://doi.org/10.17632/d8tjdxyjcm.1 | Mendeley Data, 10.17632/d8tjdxyjcm.2 |

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
