## [Editor Report]

Cholinergic projections from the basal forebrain throughout the cortex are known to regulate arousal, signal transmission and plasticity. The basal forebrain shows distinct connectivity with the cortex along its anteroposterior axis that could entail distinct modulation of different parts of cortex. By performing long-term imaging of anterior and posterior basal forebrain activity during reinforcement learning, this study finds distinct reward, learning and sensory correlates of anterior and posterior basal forebrain activity, demonstrating that the cholinergic modulation of downstream cortical areas exhibits a functional topography.

---

## [Decision Letter]

**Decision letter after peer review:**

Thank you for submitting your article "A functional topography within the cholinergic basal forebrain for processing sensory cues associated with reward and punishment" for consideration by *eLife*. Your article has been reviewed by 3 peer reviewers, and the evaluation has been overseen by Martin Vinck as the Reviewing Editor and Laura Colgin as the Senior Editor. The reviewers have opted to remain anonymous.

Essential revisions:

1) An important confound that needs to be more explicitly ruled out is the inadvertent imaging of striatal cholinergic interneurons. Specifically, if the basal ganglia structures near the GP/SI imaging site contribute to the signal, this could artificially inflate the functional differences in this nucleus compared to the HDB. This concern is particularly relevant for the photometry data, which lack cellular resolution. The results from auditory cortical GRAB sensor imaging help (Figures 1H-J), but the lack of a comparison with cortical recordings from more frontal areas makes those experiments inconclusive. In the discussion, the authors do mention lack of GCaMP expression in cholinergic striatal interneurons. Can you show quantification of this?

2) Related to point (1), we suggest showing quantification of fiber lesion locations for all animals. This is particularly important because anatomical heterogeneity is key to the findings.

3) Regarding the findings in Figures 2E-G showing very fast (1-trial) adaptation of sensory responses, another confound should be ruled out. Given that the response is higher only in the very first stimulus presentation, an alternative explanation is that, rather than reflecting subsequent adaptation, this instead reflects a startle response to that stimulus. This would be compatible with the accompanying pupil diameter data. Can the authors exclude that possibility? Is there any evidence of a startle response, for instance in body posture?

4) Discussion, ca. line 420. There was disagreement about the rationale for fiber photometry vs. microendoscopic imaging, in particular regarding lesions to the internal capsule. The authors used a fiber diameter of 400 um, which is not much smaller than a 500-um GRIN lens. Moreover, the single-plane limitation they mention could be easily circumvented with volumetric imaging using e.g. an ETL in a two-photon microscope. This point may not invalidate any of the findings, but the arguments may be inaccurate and if so, the recommendation would therefore be to remove them altogether. A key advantage of the fiber photometry approach, which the authors did not mention, is the ability to simultaneously track bulk activity across two different basal forebrain nuclei. This would be a lot more challenging with cellular-resolution approaches.

5) The authors interpret their findings on learning-related enhancement in response to reward as negative for both HDB and GP/SI. The major conclusion drawn is that HDB and GP/ SI cholinergic neurons are distinct in some of their functional roles and similar in others. It appears that distinctions between HDB and GP/ SI are not as pronounced as one might have expected. This could be due to a number of factors that may warrant deeper consideration and that need to be discussed:

a. the measured signal has relatively low cellular resolution and slow kinetics (effects of faster and/or more discrete events within the pooled signal).

b. the possible role of (poly?) synaptic interconnections between HDB and GP/ SI.

c. how does HDB vs GP/ SI Ca signaling activity relate to ACh release, behavioral output and/or engagement of their target domains (ie prefrontal cortex vs auditory cortex). In sum, how do the subtle differences in calcium dynamics between HDB and GP/SI cholinergic neurons functionally relate to the behavior of the animal?

6) While the conclusions as stated by the authors are mostly supported by the data, the conclusion that there are no reward-related learning responses in either BFCN region requires further substantiation. The authors do note that their findings are not necessarily at odds with a previous demonstration of reward-learning related enhancement within the cholinergic basal forebrain because prior studies focused on anatomically distinct populations of BFCNs (i.e. more rostral NBM and GP/SI neurons that project to the amygdala compared to caudal GP/SI neurons that project to the auditory cortex). There may be other important differences: in Crouse et al. (2020) learning-related enhancement emerged from comparisons of calcium activity in cholinergic axons in the BLA between pre-training, training, and post-training/acquisition periods. The authors in this study do not show data from pre-training (i.e. days 5-7 Figure 2A- operant shaping). The negative result of the current study would be further substantiated if the authors were to show that there are no enhancements as the mice initially learn to associate tones with rewards during this period. In the absence of such evidence, we recommend softening the conclusion that there is no reward-learning-related enhancement.

7) An additional note: upon examining heatmaps shown in Figure 3D and E for hit trials, it appears that in later trials -while there isn't an enhancement to tone responses-, there is a decrease in activity following the reward predictive tone-related activity, which isn't apparent during early trials or on miss trials (~3 seconds following tone onset). The authors should comment on whether this decrease was statistically significant.

8) Findings in this study emphasize functional heterogeneity within the basal forebrain cholinergic system. While these data are illuminating, they also present an intriguing challenge in interpreting cholinergic responses to auditory cues. If cortically projecting cholinergic neurons show activity in response to naïve auditory stimuli, as well as to stimuli that become paired with aversive outcomes with the only distinction being in the enhancement of activity upon tone-shock pairing, then how do cortical circuits interpret this potential "threshold" of cholinergic activity to gate plasticity that has been shown to occur particularly in the auditory cortex during Pavlovian aversive conditioning?

9) There are other interpretations of the data that should be considered. Since the authors are measuring the pooled activity of several neurons in each region ( i.e. fiber photometric assays of Ca signaling in the 2 BFCN regions), one might propose that the enhanced responses are a consequence of "learning-activated" privileged ensembles of neurons that are otherwise silent. Similarly, the lack of reward-related responses could also be interpreted as turning on and off different populations of neurons that could also be antagonistically connected to each other. The authors should discuss these data in more depth in the context of a "non-monolithic" basal-forebrain cholinergic system.

10) One of the major strengths of this study is that all measurements were performed in the same animal across time and regions. This strength should be further leveraged by providing a plot of z-scored dF/F responses to the auditory stimuli across the full timeline shown in Figure 2A for individual animals connected across time points to allow reviewers and readers alike to observe any differences between mice.

Of course, the strength of the repeated measures aspect of the study depends both on the precise placement and stability of the fiber optics. For completeness and especially because differences in amplitude (or lack thereof) are important to several of the conclusions, the reviewers would like to see the data on the post hoc analysis of the fiber locations in both areas for all 11 mice included in the study. The relocalization data of the fiber tips should be shown in a Supplemental figure.

Another argument in support of the assumed stability of the recording locations could be obtained if the pupil dilation x Ca signaling (as in Figure 1 E) is shown for the same optical fibers immediately post-implantation, and at 2, 3, and 4 weeks of the experiment. The authors mention that interference between their HDB and GP/SI recordings is unlikely given the different projection patterns and distances between the populations. However, there is a significant basal-forebrain to basal-forebrain connectivity within the cholinergic system (Gielow and Zaborszky 2017). The authors should discuss how this connectivity could affect their interpretations, both in terms of how one region might affect the activity in the other, and in terms of the possible contribution of axonal calcium signals from other cholinergic neurons.

11) Were the mice head-fixed on a treadmill or not? This should be clearly specified and justified, as treadmill data would have been useful for comparing with previous results on brain state changes during running vs. quiet wakefulness. Running/locomotion is associated with large changes in pupil diameter and is qualitatively different from smaller fluctuations in pupil during quiet wakefulness. It would be useful to dissociate these two conditions, and it still may be possible based on the pupil data alone. This may be an important additional analysis to clarify the role of arousal in modulating cholinergic activity.

12) Because the GCaMP signal is not deconvolved and decays slowly (especially for example after running periods), it may give the appearance of sustained or elevated cholinergic activity even in the absence of any underlying spiking. Is this what is happening in Figure 3F, G, and H in the increased baseline prior to misses. Would it be possible to do some analysis with a deconvolved signal to try to address this issue, or address this issue in another way?

13) Was there any difference in cholinergic responses to vertical vs horizontal gratings? Specific optic flow directions may be more salient/threatening to the mouse. For the interpretation of visual responses, it would be useful to dissociate this.

14) Given that the simultaneous recordings are a major strength of the study, it was somewhat disappointing that there was not more analysis of trial-to-trial variance across the two areas. Most of the results presented are averaged responses split into hits and misses. This rich data could support interesting analyses of the coordination between the two areas. The authors should consider or discuss this possible analysis.

---

## [Author Response]

Essential revisions:1) An important confound that needs to be more explicitly ruled out is the inadvertent imaging of striatal cholinergic interneurons. Specifically, if the basal ganglia structures near the GP/SI imaging site contribute to the signal, this could artificially inflate the functional differences in this nucleus compared to the HDB. This concern is particularly relevant for the photometry data, which lack cellular resolution. The results from auditory cortical GRAB sensor imaging help (Figures 1H-J), but the lack of a comparison with cortical recordings from more frontal areas makes those experiments inconclusive. In the discussion, the authors do mention lack of GCaMP expression in cholinergic striatal interneurons. Can you show quantification of this?

Thank you for raising this point. This was also a concern for us at the outset of the project, but we determined that fluorescence from the striatum was not an issue. However, we appreciate that we did not provide the evidence for this conclusion, so we have made several changes in the revised manuscript to address this important point.

1. Yes, as the reviewers suggested, the main reason is that for whatever reason, very few striatal cholinergic interneurons expressed GCaMP in the δ-neo ChAT-cre x Ai148 cross. Specifically, we counted 655 ChAT+ neurons that expressed GCaMP in GP/SI compared to just 39 neurons in the entire caudate putamen (a far larger structure) from the same sections in each of the same mice used for histology quantification. So, there are approximately 17x more GCaMP expressing cholinergic neurons in GP/SI than in neighboring regions of the caudate putamen. The same trend was observed in HDB (12x more than in rostral caudate putamen), though the HDB is not really a concern, as the reviewers noted, given its orientation relative to the fiber tip.

2. With fiber photometry, neurons within the cone of light contribute to the measured signal. Depending on the numerical aperture and fiber diameter, the bulk of the fiber photometry signal arises from fluorescence approximately 0.2mm below the fiber tip (Pisanello et al., 2019). As the reviewers can appreciate from a new figure added at their request, Figure 1 —figure supplement 1, this point is medial and ventral to the bulk of the striatum in all 11 mice.

At the reviewers’ request, the revised manuscript now includes quantification of striatal ChAT and GCaMP expression in the rostral and caudal sections (Figure 1D). We also revised the text in the beginning of the Results section to make this point clear to the reader at the outset rather than only including this point in the Discussion (Lns 153-160).

“As identified in prior studies, we observed aberrant expression in brain regions outside of the basal forebrain, including both the near-complete absence of GCaMP expression in ChAT+ striatal interneurons (Figure 1D, right) but also ectopic expression of GCaMP in ChAT-negative cells in neocortex and hippocampus. Therefore, while our transgenic strategy was appropriate for bulk imaging from cholinergic neurons in HDB and GP/SI cholinergic neurons (and in fact was aided by the absence of striatal GCaMP expression), it would not necessarily be a valid strategy for the study of other brain regions.”

2) Related to point (1), we suggest showing quantification of fiber lesion locations for all animals. This is particularly important because anatomical heterogeneity is key to the findings.

Figure 1 —figure supplement 1 addresses this suggestion by providing the location of all 22 fiber tips (2 fibers in each of 11 mice).

3) Regarding the findings in Figures 2E-G showing very fast (1-trial) adaptation of sensory responses, another confound should be ruled out. Given that the response is higher only in the very first stimulus presentation, an alternative explanation is that, rather than reflecting subsequent adaptation, this instead reflects a startle response to that stimulus. This would be compatible with the accompanying pupil diameter data. Can the authors exclude that possibility? Is there any evidence of a startle response, for instance in body posture?

This seems unlikely. Acoustic startle reflex thresholds are much higher, generally around 90 dB SPL. The highest sound level used for the spectrotemporal ripple stimuli in Figure 2 was 70 dB SPL (i.e., 10% of the signal amplitude of 90 dB), which is a moderate intensity sound that does not elicit an acoustic startle response. To address the reviewers’ comment, we included a new analysis, presented as Figure 2, Figure Supplement 2, which shows the BFCN response adaptation for visual stimuli at various contrasts and auditory stimuli at various sound levels. Although we did not measure body posture in our experiments, the reviewers will note the rapid habituation for 50 and 70 dB SPL pure tones and even low-contrast visual gratings, none of which would be expected to elicit a startle response.

Further evidence arguing against this possibility can be found in Figure 7D-G, which provides within-session changes in tone-evoked response amplitudes during the initial characterization day as well as subsequent test days. Within session adaptation to pure tones: (1) shows very different levels of adaptation between HDB and GP/SI fibers; (2) shows strong GP/SI adaptation on the initial test day but then far less adaptation on subsequent test days; (3) shows a recovery of within-session adaptation but only for the punishment-predicting cue and only in GP/SI. Each of these observations argues against the interpretation that the within-session habituation is confounded by a startle response.

As a final point, it is not clear to us that a startle response – were it to occur – would even impact our measurement. With our preparation, where the head is completely immobilized, and we are measuring bulk fluorescence with an implanted fiber. Instead, basal forebrain cholinergic neurons (BFCNs) are known to encode surprising and novel environmental stimuli, so the most likely explanation is that the response habituation reflects the rapid change in salience to a novel stimulus that is not associated with behaviorally relevant environmental consequences. This interpretation is supported by Figure 2E, where we show that sound-evoked pupil dilations are also substantially reduced after the first stimulus presentation and habituation at a similar rate to BFCN responses.

4) Discussion, ca. line 420. There was disagreement about the rationale for fiber photometry vs. microendoscopic imaging, in particular regarding lesions to the internal capsule. The authors used a fiber diameter of 400 um, which is not much smaller than a 500-um GRIN lens. Moreover, the single-plane limitation they mention could be easily circumvented with volumetric imaging using e.g. an ETL in a two-photon microscope. This point may not invalidate any of the findings, but the arguments may be inaccurate and if so, the recommendation would therefore be to remove them altogether. A key advantage of the fiber photometry approach, which the authors did not mention, is the ability to simultaneously track bulk activity across two different basal forebrain nuclei. This would be a lot more challenging with cellular-resolution approaches.

We removed the phrase from the original manuscript that the reviewers objected to, “and the large diameter would impact the internal capsule” and added the phrase, “particularly for the goal of performing simultaneous measurements of rostral and caudal BFCNs over an extended period” (Ln. 512-513). We understand the reviewer’s point regarding the challenge of recording from ensembles of neurons that are not in a single focal plane. We have toned down the language so that Ln 515-516 now states “This anatomy is not optimal for endoscopic imaging through implanted lenses, as it could be challenging to visualize BFCNs in a single focal plane.”.

5) The authors interpret their findings on learning-related enhancement in response to reward as negative for both HDB and GP/SI. The major conclusion drawn is that HDB and GP/ SI cholinergic neurons are distinct in some of their functional roles and similar in others. It appears that distinctions between HDB and GP/ SI are not as pronounced as one might have expected. This could be due to a number of factors that may warrant deeper consideration and that need to be discussed:

After reading and discussing this comment many times, we tried to come up with the most thoughtful response that we could. We have made the argument that the distinctions between the HDB and GP/SI are *more* pronounced than expected and indeed the many significant differences between these regions is the main thrust of the paper. However, we take the reviewers’ point that we still could be underestimating the degree of heterogeneity between brain structures due to some of the factors mentioned below. In regard to changes in sound-evoked responses that are associated with reward, we refer the reviewers to point #6 below where we explain changes we have made to soften our interpretation of this finding.

a. The measured signal has relatively low cellular resolution and slow kinetics (effects of faster and/or more discrete events within the pooled signal).

We explicitly mention limitations in spatial and temporal resolution with fiber-based GCaMP imaging on Lns 535-538.

“Another caveat in the interpretation of fiber-based GCaMP imaging is that the slow temporal kinetics and poor spatial resolution combines somatic and neuropil-based calcium signals and obscures the relationship to spike rates in distinct types of BFCNs.”

b. The possible role of (poly?) synaptic interconnections between HDB and GP/ SI.

The reviewers also raise this issue in Point #10, below. We have revised the text on Lns 538-549 to discuss the role of inter-regional connections between the HDB and GP/SI.

“This would be particularly worrisome if the axons of BFCNs in HDB or GP/SI projected to or through the other region, as this could either produce optical cross-talk (i.e., axon fluorescence originating from BFCNs in region A measured on the region B fiber) or functional cross-talk (i.e., projections from BFCNs in region A modulate the activity of region B). Neither of these possibilities is likely a concern in the interpretation of these findings. Correlating all single trial tone-evoked response amplitudes measured on each fiber reveals a very weak association (R^2^ = 0.16, n = 21,099 trials), demonstrating that the activity in HDB and GP/SI can be measured independently. Further, anatomical characterizations suggest that BFCN inputs within the basal forebrain primarily arise from local neurons rather than remote regions (Gielow and Zaborszky, 2017). To this point, direct visualization of efferent HDB axons showed that they left the basal forebrain in a medial and dorsal orientation, coming nowhere near the GP/SI fiber (Bloem et al., 2014).”

c. How does HDB vs GP/ SI Ca signaling activity relate to ACh release, behavioral output and/or engagement of their target domains (ie prefrontal cortex vs auditory cortex). In sum, how do the subtle differences in calcium dynamics between HDB and GP/SI cholinergic neurons functionally relate to the behavior of the animal?

We do not disagree that combining BFCN measurements with measurements of ACh release in the prefrontal cortex and other downstream structures over the course of learning, with changes in reward expectation, with changes in reward value, with changes in attention, etc. are also interesting and important studies to perform. On the other hand, we did provide one analysis of how calcium dynamics relates to cortical ACh release (Figure 1H-J) and added a new section to the Discussion on the point of how bulk calcium measurements for BFNCs could relate to downstream ACh release (Lns 457-458), “Given that GP/SI is the predominant source of BFCN input to lateral neocortical regions including ACtx, one clear implication that would be important to test in future studies is that auditory stimuli – even sounds with no behavioral relevance – should elicit ACh release in ACtx.”

In reference to the other items listed by the reviewers, we do describe BFCN responses to movement (Figure 5), reward, reward omission, punishment, and changes in punishment-predicting cues over learning (Figures 6-7). Certainly, this goes further than most any other study, particularly in terms of what has been performed in the same animal. Further to this point, we invest a great deal of effort into characterizing the association between changes in calcium signaling and the behavior of the animal. Despite all the aforementioned limitations of single-pixel bulk calcium imaging, the new main figure (Figure 4) shows that we can classify with 80% accuracy whether or not a sensory stimulus on a single trial will be subliminal or behaviorally detected based on BFCN activity preceding stimulus onset. Although the differences that distinguish hit and miss trials are *not* “subtle” (they differ by ~1 s.d.), it seems to us that they provide one important example of how “calcium dynamics between HDB and GP/SI cholinergic neurons functionally relate to the behavior of the animal”.

6) While the conclusions as stated by the authors are mostly supported by the data, the conclusion that there are no reward-related learning responses in either BFCN region requires further substantiation. The authors do note that their findings are not necessarily at odds with a previous demonstration of reward-learning related enhancement within the cholinergic basal forebrain because prior studies focused on anatomically distinct populations of BFCNs (i.e. more rostral NBM and GP/SI neurons that project to the amygdala compared to caudal GP/SI neurons that project to the auditory cortex). There may be other important differences: in Crouse et al. (2020) learning-related enhancement emerged from comparisons of calcium activity in cholinergic axons in the BLA between pre-training, training, and post-training/acquisition periods. The authors in this study do not show data from pre-training (i.e. days 5-7 Figure 2A- operant shaping). The negative result of the current study would be further substantiated if the authors were to show that there are no enhancements as the mice initially learn to associate tones with rewards during this period. In the absence of such evidence, we recommend softening the conclusion that there is no reward-learning-related enhancement.

The reviewers are correct. Thanks for pointing this out. We unplugged the mice for approximately one week in between test session #2 (characterization of sensory response habituation) and the first test session when the mice were performing the operant task. We updated the timeline in Figure 2A to make this point more explicit to the reader. During this approximate one-week period, mice were placed on water restriction and shaped on procedural demands of the Go-NoGo task that required them to produce a lick bout (≥7 licks in 2.8s) to receive water reward. During this brief shaping period, tones were associated with reward, though we did not collect imaging data because the BFCN activity related to the sensory response, licking, and reward were difficult to separate in time, hence we did not feel that it would provide an appropriate baseline measurement.

Nor did we feel that the tone-evoked responses from the sensory characterization day would be an appropriate baseline. The sensory characterization day featured many fewer pure tone presentation trials (20 trials per frequency on the test day versus ~130 on a typical operant session), were separated from the first operant session by one week, and showed strong habituation over the relatively small number of trials, all of which complicate its use as the pre-learning baseline. For these reasons, the first testing session is the most appropriate basis of comparison for changes that accompany appetitive conditioning. The reviewers will appreciate from Figure 3B-C that mice are still learning the stimulus-reward association when we resume fiber imaging at the start of the operant task.

However, the reviewers are correct that the reward outcome association was already learned by the first operant testing session, so it has its challenges as well in terms of a baseline against which to measure learning-related changes.

To address this point, we have:

1) Added Figure 7D, in which we show the tone-evoked responses during the initial sensory characterization session versus all operant sessions.

2) Edited the timeline in Figure 2A to clarify for future readers that we did not perform fiber imaging during the initial behavioral conditioning sessions.

3) We softened the conclusions throughout the paper relating to changes (or lack thereof) for sounds with learned associations to reward.

4) We explicit state the limitation in our design in identifying a true baseline related to the explanation above on Lns 267-273:

“Another possibility is that response enhancement to reward-predicting sounds had already occurred during the initial shaping period that preceded the first operant imaging session, thereby escaping our analysis. Although performance in the Go-NoGo auditory task clearly improved over the course of our imaging period (Figure 3B-C), learning related enhancements of cue-evoked BFCN responses can occur within just a few behavioral sessions (Crouse et al., 2020; Sturgill et al., 2020), so we cannot rule out this possibility.”

And lines 486-489:

“As mentioned above, another possibility is that HDB and GP/SI BFCNs did exhibit an increased response to reward-predicting cues during the initial association of sound and reward, which occurred during the behavioral shaping period when we did not monitor activity.”

7) An additional note: upon examining heatmaps shown in Figure 3D and E for hit trials, it appears that in later trials -while there isn't an enhancement to tone responses-, there is a decrease in activity following the reward predictive tone-related activity, which isn't apparent during early trials or on miss trials (~3 seconds following tone onset). The authors should comment on whether this decrease was statistically significant.

The reviewers are referring to the bottom-left sub-panels of Figures 3F and 3G, which shows a slightly elevated mean response 3-6s post-stimulus onset early in training compared to late in training. The difference is statistically significant, but we did not choose to emphasize it in the manuscript because the effect is confounded by differences in lick rate. We have created Figure 3 —figure supplement 1 to address the reviewers’ point.

This figure shows that our mice licked the reward spout for a longer duration earlier in training. Because the lick rate is also elevated in the 3-6s post-stimulus period, the most likely explanation is that the elevated sustained activity reflects the difference in motor activity (which we examine directly in Figure 5). As further proof, we can also look at lick rate in the Miss trials. Later in training, mice tend to lick the spout several seconds after the stimulus was presented (as if they heard it but were too slow to respond). On miss trials, the BFCN activity in the same 3-6s post-stimulus period is also elevated, except here in the *opposite* direction, with activity higher late in training compared to early. In both cases, the elevated activity at longer intervals after stimulus offset may be explained by systematic differences in lick rate. We have edited Lns 260-262 of the text to clarify this point.:

“Responses were slightly elevated at longer latencies after stimulus onset early in training, though this difference could be explained by differences in lick rate duration over the course of training (Figure 3 —figure supplement 1).”

8) Findings in this study emphasize functional heterogeneity within the basal forebrain cholinergic system. While these data are illuminating, they also present an intriguing challenge in interpreting cholinergic responses to auditory cues. If cortically projecting cholinergic neurons show activity in response to naïve auditory stimuli, as well as to stimuli that become paired with aversive outcomes with the only distinction being in the enhancement of activity upon tone-shock pairing, then how do cortical circuits interpret this potential "threshold" of cholinergic activity to gate plasticity that has been shown to occur particularly in the auditory cortex during Pavlovian aversive conditioning?

Yes, exactly. This is one of the most intriguing and surprising findings that have emerged from this study and the prior study from our lab because it challenges the interpretation that ACh acts as a simple gating mechanism (i.e., on/off) that enables associative plasticity to enhance the representation of behaviorally relevant stimuli. We have now confirmed this result with many different approaches: (1) by recording from optogenetically isolated BFCN units in GP/SI that project to ACtx (Guo et al., Neuron, 2019), by measuring bulk GCaMP activity in GP/SI (this manuscript), and even ACh3.0 sensor activity in several cortical regions (unpublished). They all emphasize that same finding: that moderate intensity sounds with no explicit hedonic value elicit rapid and robust cholinergic responses. As the reviewers point out and we have shown directly in our 2019 paper, there is no long-term representational plasticity in the cortex for sounds experienced in a passive context or for sounds that do not predict reinforcement outcomes, so something beyond the mere presence of ACh must be required to create a permissive state in the cortex for long-term plasticity. Our 2019 paper and a recent paper from the Hangya lab (Laszlovszky et al., Nat. Neuro., 2020) both made paired recordings from BFCN and ACtx units during reinforced auditory learning versus passive listening and both studies identified marked upticks in the LFP coherence and spike-triggered LFP when animals process sounds in the context of Pavlovian auditory trace conditioning (Guo et al., 2019) or operant auditory conditioning (Laszlovszky et al., Nat. Neuro., 2020).

Although the underlying mechanism is not yet clear, these changes in inter-regional coherence were only observed from BFCN recordings, not when other putative non-cholinergic basal forebrain units were recorded, suggesting that the critical factor that switches downstream networks into modes of plasticity is still related to basal forebrain cholinergic activity, just not the presence or absence of activity. As our study and others (e.g., Crouse et al., *eLife*, 2020) have noted, the BFCN response is itself plastic, and increases for sounds paired with behavioral reinforcement. As the reviewers suggested, this raises the possibility that the permissive feature is a threshold of ACh activity that is only reached during reinforced learning. Another possibility is that the downstream plasticity is enabled by a confluence of cholinergic inputs from the caudal tail of the basal forebrain (where BFCNs respond even to unconditioned neutral sounds) and more rostral regions (e.g., nucleus basalis or portions of SI) where sensory responses are acquired during learning. These hypotheses (and others) can be tested in future studies that combine regionally targeted BFCN recordings with optogenetic silencing and cholinergic sensory measurements.

We refer the reviewers to Lns 457-475 of the Discussion where we address the reviewers’ points directly in an abbreviated form:

“Given that GP/SI is the predominant source of BFCN input to lateral neocortical regions including ACtx, one clear implication that would be important to test in future studies is that auditory stimuli – even sounds with no behavioral relevance – should elicit ACh release in ACtx. ACh acts through local ACtx microcircuits to remove the fetters that normally limit long-term associative plasticity, thereby enabling local synaptic processes that support auditory fear memory encoding (Letzkus et al., 2015; Weinberger, 2004) and perceptual learning (Froemke et al., 2013; Takesian et al., 2018). Importantly, learning-related plasticity in ACtx requires transient neuromodulatory surges and does not occur when stimuli are presented in a passive context (Froemke, 2015). This suggests cholinergic regulation of cortical plasticity is not an all-or-none gating process but instead may reflect a threshold that is only exceeded when sound-evoked cholinergic inputs are themselves transiently amplified through learning (Figure 7C and 7G, Guo et al., 2019). Beyond simple gating mechanisms, dual recordings from single BFCNs and ACtx neurons during Pavlovian auditory learning (Guo et al., 2019) and attentionally demanding auditory tasks (Laszlovszky et al., 2020) have demonstrated dynamics in BFCN-cortex synchrony that change in lockstep with associative plasticity and auditory perceptual salience. Although the upstream factors that regulate BFCN plasticity and inter-regional synchrony have yet to be identified, it is clear that models portraying phasic cortical ACh release occurring only at times of reward, punishment, or heightened arousal need to be reevaluated, at least as they relate to the caudal tail of the basal forebrain and the ACtx.”

9) There are other interpretations of the data that should be considered. Since the authors are measuring the pooled activity of several neurons in each region (i.e. fiber photometric assays of Ca signaling in the 2 BFCN regions), one might propose that the enhanced responses are a consequence of "learning-activated" privileged ensembles of neurons that are otherwise silent. Similarly, the lack of reward-related responses could also be interpreted as turning on and off different populations of neurons that could also be antagonistically connected to each other. The authors should discuss these data in more depth in the context of a "non-monolithic" basal-forebrain cholinergic system.

The reviewers are correct; these are important factors to consider. We now explicitly mention both of these interpretations in the revised “Technical considerations in the interpretation of these findings” section on Lns. 527-535.

“However, there are important limitations and technical caveats with fiber-based bulk GCaMP imaging that should be taken into consideration in the interpretation of these findings. Because fiber photometry signals arise from populations of neurons, it is impossible to discern whether differences in response amplitude over learning or across different behavioral states reflect the activation of privileged ensembles that were hitherto silent or instead an increased response expressed uniformly across neurons. Conversely, the absence of differences in the population signal could belie striking shifts in the representational dominance of antagonistically related cellular ensembles that would not be captured by changes in net signal amplitude (Grewe et al., 2017; Gründemann, 2021; Taylor et al., 2021).”

10) One of the major strengths of this study is that all measurements were performed in the same animal across time and regions. This strength should be further leveraged by providing a plot of z-scored dF/F responses to the auditory stimuli across the full timeline shown in Figure 2A for individual animals connected across time points to allow reviewers and readers alike to observe any differences between mice.

Thanks for this suggestion. We created a new figure, Figure 7D, which shows tone-evoked response amplitude across all measurement sessions and regions within single animals, as requested. This figure proves useful for addressing several other points above and below, so we thank the reviewers for making this suggestion.

Of course, the strength of the repeated measures aspect of the study depends both on the precise placement and stability of the fiber optics. For completeness and especially because differences in amplitude (or lack thereof) are important to several of the conclusions, the reviewers would like to see the data on the post hoc analysis of the fiber locations in both areas for all 11 mice included in the study. The relocalization data of the fiber tips should be shown in a Supplemental figure.

Another good suggestion, thanks. We created a new figure, Figure 1 —figure supplement 1 that provides the post-hoc localization of the fiber center and diameter for all 22 fibers from 11 mice.

Another argument in support of the assumed stability of the recording locations could be obtained if the pupil dilation x Ca signaling (as in Figure 1 E) is shown for the same optical fibers immediately post-implantation, and at 2, 3, and 4 weeks of the experiment.

Figure 7C-D shows tone-evoked response amplitudes from the same fiber over many weeks, which should be sufficient to address the possibility that the implanted fiber tip is not stable. As evidenced by this figure and all of the other analyses in the paper that show sensory-evoked bulk BFCN GCaMP responses over time, if the fiber tip were moving away from GCaMP-expressing cells over time, the signal amplitude would drop out or change indiscriminately across all acoustic test frequencies. Instead, we report that changes in tone-evoked response amplitude either don’t occur (HDB) or occur only for tones associated with punishment (GP/SI), which rules out that possibility. It is not clear why the reviewers felt that pupil-related changes would be a better demonstration for mechanical stability, but it ended up being a moot point for our data because lighting and camera position was not optimized for pupil measurements after the initial test day. These findings argue against the possibility that there were gross changes in the recording location over time. Of course, we cannot rule the possibility that there was some degree of mechanical drift over time that would change the contribution of some neurons within the cone of light across measurements, but it is not clear why the reviewers think this is a pressing issue for an approach that cements a 0.4mm optical fiber into a deep brain area as compared to approaches for long-term chronic neural monitoring based on very thin flexible wires. If anything, mechanical drift should be less of a concern.

The authors mention that interference between their HDB and GP/SI recordings is unlikely given the different projection patterns and distances between the populations. However, there is a significant basal-forebrain to basal-forebrain connectivity within the cholinergic system (Gielow and Zaborszky 2017). The authors should discuss how this connectivity could affect their interpretations, both in terms of how one region might affect the activity in the other, and in terms of the possible contribution of axonal calcium signals from other cholinergic neurons.

Thanks, we address this point explicitly in the revised Discussion. One way to address the reviewers’ point about functional or even direct optical (viz. inter-regional axon projections) influences between fibers is simply to correlate the responses between the two fibers. We did this for every trial of every tone burst presented to every mouse (n = 21,099 trials, see Author response image 1). Even though some degree of correlation is expected from common inputs to both regions (e.g., more intense stimuli elicit more intense responses, responses on hit trials are higher than miss trials) a linear model can only account for 16% of the variance in response amplitude for one fiber knowing the amplitude of the other. We feel that this constitutes fairly strong evidence against the hypothesis that there may be substantive functional/optical cross-talk between our fibers. Generally, the concern about cross-talk is greater for conclusions that anatomically separate regions of the BFCN responded uniformly to various experimental conditions. The main point throughout the paper is just the opposite.

**Author response image 1. sa2fig1:** 

We believe the reviewers may be mistaken in their interpretation of the Gielow and Zaborsky paper. In this study, the authors injected the starter virus throughout the rostral-caudal length of the basal forebrain, so they would not be able to make a direct measurement of long-range connections between regions. Further, the input cells labeled with the pseudo-typed rabies virus would not necessarily be cholinergic neurons, which further obscures the ability to make any direct assessments of long-range projections between cholinergic neurons. Despite these caveats, the authors conclude on pg. 1825 of the published paper “Across all cytoarchitectonic BFc subregions in all subjects, the number of labeled afferents within a BFc subdivision correlates significantly with the number of starter cells within the same structure (Spearman’s Rho = 0.77; p < 0.001), suggesting that cholinergic corticopetal neurons likely receive local inputs primarily from neighboring rather than remote regions of the BF itself.” In direct personal communication with Prof. Zaborsky about this point, he confirmed that his paper does not make claim “significant basal-forebrain to basal-forebrain connectivity within the cholinergic system” and, if anything, makes the opposite point. The experiment that would test the reviewers’ suggestion would have to restrict the expression of starter virus to cholinergic neurons in one region of the basal forebrain and then co-label the input neurons with ChAT. To the best of our knowledge, there is no public record of this experiment having been performed. In the absence of this direct experiment, the best evidence we can find comes from descriptions of the efferent projection patterns from cholinergic neurons in focal regions of the basal forebrain. As cited in our manuscript, there is no description of efferent axons from focal regions of the basal forebrain traveling through the length of the basal forebrain, rather they exit in distinct trajectories through major white matter fiber bundles.

These points are summarized on Lns 538-549, where we state:

“This would be particularly worrisome if the axons of BFCNs in HDB or GP/SI projected to or through the other region, as this could either produce optical cross-talk (i.e., axon fluorescence originating from BFCNs in region A measured on the region B fiber) or functional cross-talk (i.e., projections from BFCNs in region A modulate the activity of region B). Neither of these possibilities is likely a concern in the interpretation of these findings. Correlating all single trial tone-evoked response amplitudes measured on each fiber reveals a very weak association (R^2^ = 0.16, n = 21,099 trials), demonstrating that the activity in HDB and GP/SI can be measured independently. Further, anatomical characterizations suggest that BFCN inputs within the basal forebrain primarily arise from local neurons rather than remote regions (Gielow and Zaborszky, 2017). To this point, direct visualization of efferent HDB axons showed that they left the basal forebrain in a medial and dorsal orientation, coming nowhere near the GP/SI fiber (Bloem et al., 2014).”

11) Were the mice head-fixed on a treadmill or not? This should be clearly specified and justified, as treadmill data would have been useful for comparing with previous results on brain state changes during running vs. quiet wakefulness. Running/locomotion is associated with large changes in pupil diameter and is qualitatively different from smaller fluctuations in pupil during quiet wakefulness. It would be useful to dissociate these two conditions, and it still may be possible based on the pupil data alone. This may be an important additional analysis to clarify the role of arousal in modulating cholinergic activity.

No, the mice were not on a treadmill. This has now been clarified in the Materials and methods (Ln 633): “Mice were placed in an electrically conductive cradle…” and on lns 665-666 “On sessions 1 and 2, mice were habituated to head fixation and the body cradle.” The coherence analysis (Figure 1F-G) shows two peaks between the spontaneous pupil changes and BFCN activity, one corresponding to very slow changes (0.05 Hz) and the other corresponding to more rapid fluctuations (0.4 Hz). The coherence reported here (both between pupil and bulk BFCN activity as well as pupil and cortical ACh release [Figure 1I-J]) resembles previous descriptions of pupil and cholinergic axon activity (see Figure 1h of Reimer et al., Nat Comm 2016) and pupil and ACtx membrane potential (see Figure 2C of McGinley et al., Neuron 2015).

12) Because the GCaMP signal is not deconvolved and decays slowly (especially for example after running periods), it may give the appearance of sustained or elevated cholinergic activity even in the absence of any underlying spiking. Is this what is happening in Figure 3F, G, and H in the increased baseline prior to misses. Would it be possible to do some analysis with a deconvolved signal to try to address this issue, or address this issue in another way?

The newly added Figure 5 —figure supplement 1 is relevant to this point by demonstrating that BFCN activity is not elevated prior to false alarms events. The mice are head-fixed in a small cradle so they are definitely not running and – from spending countless hours watching them perform the task – they really don’t exhibit any type of gross motor behavior (e.g., squirming, struggling etc.) when performing the operant task. Instead, on miss trials, mice just appear to “zone out” from time to time and they fail to hear the target sounds. The newly added Figure 4 demonstrates that we can classify miss trials with ~80% accuracy from the activity preceding cue onset. Because HDB is the better predictor of trial outcome but is less modulated by movement than GP/SI, these effects are unlikely due to sustained movement.

There is no way to deconvolve a bulk GCaMP signal because there is no “ground truth” underlying spiking signal from which to create the convolution function. By contrast, in combined intracellular and GCaMP recordings in single units, one has the ground truth spiking and membrane potential measurements alongside the GCaMP, from which one can calculate a kernel to deconvolve spike rates from fractional change in fluorescence. Of course, one can always apply the single unit GCaMP deconvolution filter to bulk imaging signals but it would just be one of many forms of temporal filtering that would not accomplish the objective of accurately estimating the underlying spiking.

13) Was there any difference in cholinergic responses to vertical vs horizontal gratings? Specific optic flow directions may be more salient/threatening to the mouse. For the interpretation of visual responses, it would be useful to dissociate this.

We created a new figure to address this question (Figure 2 —figure supplement 1). There are no systematic differences in the responses to vertical vs horizontal visual gratings, nor upward vs. downward auditory gratings.

14) Given that the simultaneous recordings are a major strength of the study, it was somewhat disappointing that there was not more analysis of trial-to-trial variance across the two areas. Most of the results presented are averaged responses split into hits and misses. This rich data could support interesting analyses of the coordination between the two areas. The authors should consider or discuss this possible analysis.

Based on this suggestion, we created a new main Figure, Figure 4, that presents single trial decoding of behavioral trial outcome based on the activity of each fiber individually, or both fibers combined. This figure shows that we predict trial outcome (hit or miss) with ~80% accuracy based on the mean GCaMP activity before stimulus onset, which is even more accurate than the classification from the stimulus-evoked response itself. HDB classification accuracy was greater overall than GP/SI, and there was no statistical advantage to decoding with both brain areas simultaneously than with just HDB. We feel that this single trial analysis constitutes an important addition to the study and addresses the reviewers’ request for more analysis of trial-to-trial variability, even if it doesn’t reveal any particular cooperativity or computational advantage to decode trial outcome using both regions of the basal forebrain. We tried many other analyses as well that would exploit the advantage of having simultaneous recordings (e.g., noise and signal correlations, coherence analysis) but nothing stood out as being particularly clear or interpretable.

It is probably not accurate for the reviewers to state “most of the results presented are averaged responses split into hits and misses”. We provide side-by-side visualization of single trial data from both fibers in Figure 2B, Figure 2 FS1A, Figure 3D-E, Figure 4A and C, Figure 5A, Figure 6D, and Figure 7D. Further, hits and misses are only characterized in Figures 3 and 4, which hardly qualifies as “most”. We have made a point of showing raw data in every figure and presenting findings at the individual animal level, wherever possible (i.e, in every figure). Having two fibers also allows us to perform paired statistics rather than unpaired statistics, which reduces the influence of inter-animal variability that can be an issue for approaches that only study one brain structure per animal. Between the addition of the new classifier analysis in Figure 4 and the points above, we hope the reviewers will feel somewhat less disappointed.